# Sirtuin 6 maintains epithelial STAT6 activity to support intestinal tuft cell development and type 2 immunity

Xiwen Xiong [1,2] ✉, Chenyan Yang[1,2], Wei-Qi He [3], Jiahui Yu[1,2], Yue Xin[1,2], Xinge Zhang[1,2], Rong Huang[1,2], Honghui Ma[1,2], Shaofang Xu[3], Zun Li[1,2], Jie Ma[4], Lin Xu[5], Qunyi Wang[6], Kaiqun Ren[7], Xiaoli S. Wu[8], Christopher R. Vakoc [8], Jiateng Zhong[4], Genshen Zhong [5], Xiaofei Zhu[5], Yu Song[9], Hai-Bin Ruan [10] & Qingzhi Wang[1] ✉

Dynamic regulation of intestinal epithelial cell (IEC) differentiation is crucial for both homeostasis and the response to helminth infection. SIRT6 belongs to the NAD+-dependent deacetylases and has established diverse roles in aging, metabolism and disease. Here, we report that IEC *Sirt6* deletion leads to impaired tuft cell development and type 2 immunity in response to helminth infection, thereby resulting in compromised worm expulsion. Conversely, after helminth infection, IEC SIRT6 transgenic mice exhibit enhanced epithelial remodeling process and more efficient worm clearance. Mechanistically, *Sirt6* ablation causes elevated *Socs3* expression, and subsequently attenuated tyrosine 641 phosphorylation of STAT6 in IECs. Notably, intestinal epithelial overexpression of constitutively activated STAT6 (STAT6vt) in mice is sufficient to induce the expansion of tuft and goblet cell linage. Furthermore, epithelial STAT6vt overexpression remarkably reverses the defects in intestinal epithelial remodeling caused by *Sirt6* ablation. Our results reveal a novel function of SIRT6 in regulating intestinal epithelial remodeling and mucosal type 2 immunity in response to helminth infection.

The intestinal epithelium is a monolayer of columnar epithelial cells lining the luminal surface and functions as a physical and immunological barrier between the host and its luminal contents[1]. Two major differentiated cell types are defined within the intestinal epithelium, (a) absorptive enterocytes and (b) secretory lineages, which include Paneth cells that release antimicrobial factors, goblet cells that produce mucins, enteroendocrine cells that secrete hormones and tuft cells which play a key role in helminth expulsion[2,3].

Parasitic helminth infections are a major public health problem and affect more than 1.5 billion people worldwide[4,5]. The host defense against helminth infection is orchestrated by activation of type 2 immunity[6,7]. One of the characteristic type 2 immune responses is intestinal epithelial remodeling[8]. Although tuft and goblet cells constitute only a minor fraction of intestinal epithelial cells (IECs), tight

control of their differentiation is crucial for protection against infections with gastrointestinal parasitic worms[9,10]. Chemosensory tuft cells sense the intestinal parasitic infection and produce IL25, which causes accumulation of type 2 innate lymphoid cells (ILC2s) and other IL4/IL13 producing cell types in the lamina propria (LP). In turn IL4/IL13 signals to the stem cells within the intestinal crypt and promotes tuft and goblet cell hyperplasia[11–13]. STAT6 is a well-known transcription factor to mediate type 2 immune responses following IL4/IL13 activation, playing important roles in the clearance of intestinal parasites and allergic disorders[14,15]. In response to IL4/IL13, STAT6 proteins undergo tyrosine phosphorylation and are transported to the nucleus, where along with co-factors, STAT6 regulates the expression of its target genes[16]. Confirming the requirement for STAT6 in anti-helminth responses, mice lacking STAT6 show abrogated expansion of tuft

and goblet cells and compromised worm expulsion after helminth infections[12,17–19]. Although STAT6 has been mainly shown to function in immune cells, there is a growing appreciation of its role in non-immune cells, including IECs. Selective expression of constitutively activated STAT6 in IECs promotes proliferation and differentiation of tuft and goblet cells and protects against gastrointestinal helminth infections, indicating activation of STAT6 in epithelial cells is critical for secretory cell differentiation[20].

The Sirtuin family members (SIRT1-7) are evolutionarily conserved proteins with enzymatic activity of NAD⁺-dependent deacetylase or mono-ADP-ribosyltransferase[21]. Sirtuin 6 (SIRT6) is a nuclear and chromatin-bound protein that emerges as an important epigenetic regulator controlling longevity, genome stability, metabolism, and inflammation[22,23]. *Sirt6* systemic knockout causes severe hypoglycemia and premature death, while transgenic mice overexpressing SIRT6 have an extended lifespan in males[24–26]. Notably, IEC-specific *Sirt6* knockout mice are more susceptible to dextran sulfate sodium (DSS)-induced colitis. Importantly, the protective role of SIRT6 in colitis maybe mediated by R-spondin-1 because SIRT6 fails to prevent proinflammatory cytokines induced YAMC (a normal colonic epithelial cell line) cell death when *Rspo1* is silenced by siRNA[27]. Conversely, transgenic mice overexpressing SIRT6 exhibit improvement of DSS-induced colitis progression probably due to attenuated activation of NF-kB and c-Jun pathways in the colon[28]. However, the roles of SIRT6 in regulating intestinal secretory cell lineage development and type 2 mucosal immunity in response to helminth infections have not been studied.

In the present study, we demonstrate that epithelial SIRT6 is essential for tuft cell development and maintaining intestinal type 2 immunity in response to helminth infection. In IECs, SIRT6 inhibits SOCS3 transcription through histone H3K56 deacetylation, thereby enhancing STAT6 activity to promote intestinal epithelial remodeling for helminth expulsion. This study unveils a novel role of SIRT6 in type 2 mucosal immunity in the context of parasitic infections.

## Results

### *Sirt6* deletion in IECs leads to impaired tuft cell development and intestinal type 2 immunity

*Sirt6* floxed mice were crossed with *Villin-Cre* mice to generate IEC-specific *Sirt6* knockout mice (IEC-KO, genotype: *Sirt6^{flox/flox};VillinCre*) and control mice (LoxP, genotype: *Sirt6^{flox/flox}*). Knockout of *Sirt6* in the jejunal or colonic epithelium of IEC-KO mice was confirmed at both mRNA and protein levels (Fig. 1a, b). IEC-KO mice were born at the expected Mendelian frequencies and exhibited no overt abnormalities. In addition, H&E staining of the jejunum and colon tissues revealed no noticeable mucosal abnormalities in IEC-KO mice compared with LoxP mice (Supplementary Fig. 1a–d). To understand the functional outcome of *Sirt6* ablation in IEC differentiation, we analyzed the mRNA expression of cell lineage markers in the jejunal and colonic IECs. Interestingly, we found *Dclk1* (tuft cell marker) expression was significantly reduced in IEC-KO mice compared with LoxP littermates (Fig. 1c, d). In contrast, epithelial expression of markers for goblet cells (*Muc2*), enterocytes (*Slc5a1*), enteroendocrine cells (*Chga*), Paneth cells (*Lyz1*), and stem cells (*Lgr5*) was comparable in LoxP and IEC-KO mice (Fig. 1c, d). To test whether *Sirt6* ablation altered the epithelial cell life cycle, we further analyzed cell proliferation (Ki67) and apoptosis (TUNEL) in the jejunum but found no significant differences between LoxP and IEC-KO mice (Supplementary Fig. 1e–h).

In the small intestine (SI), tuft cells sense helminth infections and initiate a type 2 immune response characterized by epithelial remodeling. *Heligmosomoides polygyrus* (*H. poly*), which induces type 2 immune responses, is a natural intestinal dwelling parasitic helminth of mice[29,30]. Indeed, we observed noticeable tuft and goblet cell hyperplasia during infection of mice with *H. poly*, as shown by qPCR and histological analyses (Supplementary Fig. 2a–d). To further investigate the role of SIRT6 in helminth infection-induced intestinal epithelial

remodeling, we infected LoxP and IEC-KO mice with *H. poly* and analyzed on day 14 p.i., a time point at which adult worms were detected in the SI and eggs were found in the feces[31]. Both the tuft cell numbers and the expression of tuft cell marker genes including *Dclk1*, *Il25*, and *Pou2f3* were reduced in the jejunum of naive IEC-KO mice (Fig. 1e, g, k). As expected, *H. poly* infection led to dramatic tuft cell hyperplasia in the jejunum of LoxP mice but blunted tuft cell expansion in IEC-KO mice (Fig. 1e, g, k). Although the goblet cell abundance was not significantly altered in naive IEC-KO mice, we found the numbers of goblet cell and the cell markers (*Muc2*, *Retnlb*) expression were significantly reduced in the jejunum of *H. poly*-infected IEC-KO mice compared with LoxP littermates, indicating that *H. poly* infection-induced goblet cell hyperplasia is compromised when epithelial *Sirt6* is ablated (Fig. 1f, h, k). As a result of the defective epithelial remodeling, IEC-KO mice had more adult worms in the SI and passed more eggs into the feces (Fig. 1i, j).

Since intestinal tuft cell-ILC2 circuit mediates epithelial responses to helminth infections, we then performed flow cytometric analysis to determine the frequency of intestinal LP ILC2s in *H. poly*-infected LoxP and IEC-KO mice. Surprisingly, IEC-KO mice had comparable frequency of ILC2s compared with LoxP mice (Fig. 1l). Notably, *H. poly*-infected IEC-KO mice had lower mRNA expression of type 2 cytokine genes in the jejunum than LoxP mice, suggesting that LP ILC2s from IEC-KO mice maybe functionally impaired (Fig. 1m). Since IL13 released by ILC2 is critical for intestinal epithelial remodeling in response to type 2 immunity, we further explored the IL13 levels in the jejunum by ELISA. Consistently, compared with LoxP controls, IEC KO mice had reduced jejunal IL13 levels after *H. poly* infection (Fig. 1n). Collectively, these data demonstrate that SIRT6 is required for tuft cell development and intestinal anti-helminth responses.

To exclude the possibility that the epithelial abnormalities observed in IEC-KO mice were due to the side effects of Cre expression driven by *Vil1* promoter, we evaluated the jejunal anti-helminth responses in *Sirt6^{flox/+}* mice and heterozygous *Sirt6^{flox/+}:VillinCre* mice. As expected, *Sirt6^{flox/+}:VilCre* mice had no apparent changes in the numbers of tuft and goblet cells compared with *Sirt6^{flox/+}* mice (Supplementary Fig. 3a–c). Accordingly, worm burden on day 14 after *H. poly* infection was also comparable in *Sirt6^{flox/+}* and *Sirt6^{flox/+}:VilCre* mice (Supplementary Fig. 3d, e). These results confirm the specificity of SIRT6 in regulating anti-helminth responses in the intestinal epithelium.

Succinate triggers a type 2 immune responses in the SI by activating tuft cells, thereby leading to the epithelial remodeling[32,33]. To test whether SIRT6 is also required for succinate-induced tuft cell expansion, we supplemented mouse drinking water with 150 mM succinate for 7 days and measured tuft cell hyperplasia in the jejunum of LoxP and IEC-KO mice. Succinate treatment induced a significant increase in the numbers of both tuft and goblet cells in WT mice, however, *Sirt6* deletion in IECs compromised the succinate-induced tuft and goblet cell hyperplasia (Supplementary Fig. 4a–c).

### IL4&IL13 treatment is not sufficient to rescue the defective *H. poly* infection-induced tuft and goblet cell hyperplasia in IEC-KO mice

As we shown above, tuft cells, ILC2s and epithelial progenitors comprise a response circuit that mediates intestinal epithelial remodeling associated with type 2 immunity. To explore to what extent the defective type 2 immunity caused by diminished tuft cell abundance contributes to the impaired anti-helminth responses observed in IEC-KO mice, we treated *H. poly*-infected LoxP and IEC-KO mice with recombinant murine IL4/IL13 mixture for 8 days (started on 6 d.p.i.) and assessed the phenotypes of epithelial remodeling and worm clearance. Importantly, rIL4/rIL13 treatment greatly increased the resistance of mice to *H. poly* infection and largely rescued the impaired ability to expel *H. poly* worms in IEC-KO mice, as illustrated by comparable numbers of adult worms in the lumen and eggs in the feces between the two genotypes (Fig. 2a, b).

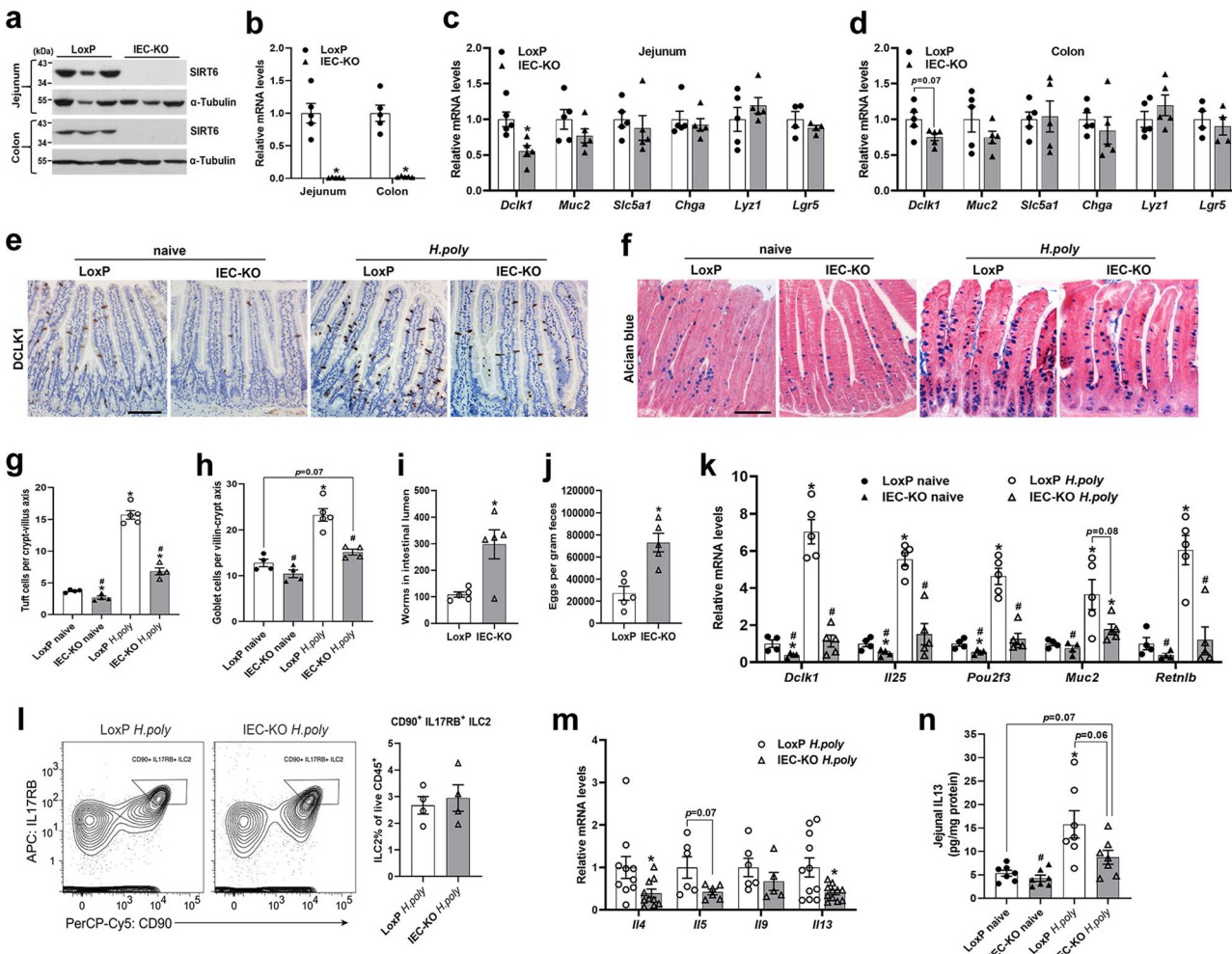

**Fig. 1 | IEC-specific deletion of *Sirt6* results in defective tuft cell expansion and intestinal type 2 immune responses.** 2–3-month-old male naive and/or *H. poly*-infected LoxP and IEC-KO mice were subjected to the following assays on day 14 post-infection. **a, b** Western blot (**a**) and qPCR (**b**) analyses of SIRT6 in the jejunal and colonic IECs of naive mice. *n* = 5 mice/group in (**b**); *p* = 0.0028 (jejunum) and 0.0014 (colon). **c, d** qPCR analysis of the expression of IEC markers in the jejunal (**c**) and colonic (**d**) IECs of naive mice. *Dclk1, Muc2, Slc5a1, Chga, Lyz1*, *n* = 5 mice/group; *Lgr5*, *n* = 4 mice/group; *p* = 0.0096 (*Dclk1*) in (**c**). **e** Jejunal tuft cells were examined by immunostaining with anti-DCLK1 antibody (200X). **f** Jejunal goblet cells were examined by Alcian blue staining (200X). **g** Quantification of tuft (DCLK1 positive) cells shown in (**e**). *n* = 4, 4, 5, 4 mice/group, respectively; 15 crypt-villus units counted for each mouse; from left to right, **p* = 0.0188, <0.0001, 0.0093; #*p* < 0.0001 for both. **h** Quantification of goblet (Alcian blue positive) cells shown in (**f**). *n* = 4, 4, 5, 4 mice/group, respectively; 15 crypt-villus units counted for each mouse; from left to right, **p* = 0.0005; #*p* = 0.0001, 0.0021. **i, j** Analysis of parasite

burden by quantification of adult worms in intestinal lumen (**i**) and eggs in feces (**j**). *n* = 5 mice/group; *p* = 0.0243 in (**i**) and 0.0029 in (**j**). **k** qPCR analysis of tuft and goblet cell markers expression in the jejunal IECs. *n* = 4, 4, 5, 5 mice/group, respectively; from left to right, *Dclk1* (**p* = 0.0497, 0.0004; #*p* = 0.0005, 0.0002); *Il25* (**p* = 0.0474, <0.0001; #*p* < 0.0001, 0.0006); *Pou2f3* (**p* = 0.032, 0.0007; #*p* = 0.0006, 0.0004); *Muc2* (**p* = 0.0296, 0.0466; #*p* = 0.0214); *Retnlb* (**p* = 0.0016; #*p* = 0.0019, 0.0018). **l** Representative FACS plots of *H. poly*-infected mice analyzed for IL17RB+CD90+ ILC2s gated from Lin−CD45+CD3−CD4− cells in the intestinal lamina propria. *n* = 4 mice/group. **m** qPCR analysis of type 2 cytokines expression in the jejunum of mice infected with *H. poly*. *n* = 10, 6, 6, 11 mice/group for *Il4, Il5, Il9, Il13*, respectively; *p* = 0.0484 (*Il4*) and 0.0269 (*Il13*). **n** Jejunal IL13 levels were measured by ELISA. *n* = 7 mice/group; **p* = 0.0114; #*p* = 0.0072. Data are presented as mean ± SEM. All *p* values were generated by two-tailed unpaired *t* test. **p* < 0.05. In (**g, h, k, n**), **p* < 0.05 vs LoxP naive, #*p* < 0.05 vs LoxP *H. poly*. Scale bars, 100 μm. Source data are provided as a Source Data file.

In contrast, rIL4/rIL13 treatment did not rescue the blunted tuft and goblet cell expansion following *H. poly* infection in IEC-KO mice (Fig. 2c–g). These data suggest that epithelial *Sirt6* ablation impairs the activity of IL4/IL13 signaling pathway in IECs and thereby inhibits the type 2 immunity-induced intestinal epithelial remodeling. STAT6, a transcription factor mediating type 2 immune responses, is found in the cell in its latent form. Upon cytokine stimulation it is phosphorylated by the Jak kinases at Tyr641, this leads to dimerization and translocation of STAT6 to the nucleus to induce transcription[14]. Indeed, we observed significantly reduced Y641 phosphorylation of STAT6 in the jejunal epithelium of *H. poly*-infected IEC-KO mice in the absence or presence of rIL4/rIL13 treatment, implying a potential role of SIRT6 in IL4/IL13-STAT6 pathway regulation in IECs (Fig. 2h, i).

## Deficiency of SIRT6 in intestinal organoids leads to compromised IL13-induced tuft and goblet cell expansion

To further explore whether SIRT6 regulates helminth infection-induced epithelial remodeling in an epithelial cell autonomous manner, we used an ex vivo organoid culture system that allows physiological responses of isolated intestinal epithelium to be analyzed in the absence of stromal cues. We set up intestinal organoid cultures from LoxP and IEC-KO mice and treated the organoids with vehicle or rIL13. *Sirt6* deletion in organoids resulted in significantly reduced mRNA expression of tuft cell markers (*Dclk1, Trpm5, Pou2f3*) in both vehicle- and rIL13-treated conditions (Fig. 3a). Consistent with the in vivo data, we also found noticeably downregulated expression of goblet cell markers (*Muc2, Retnlb*) in rIL13-treated IEC-KO organoids (Fig. 3a). To further validate our qPCR results, we stained vehicle- or rIL13-treated

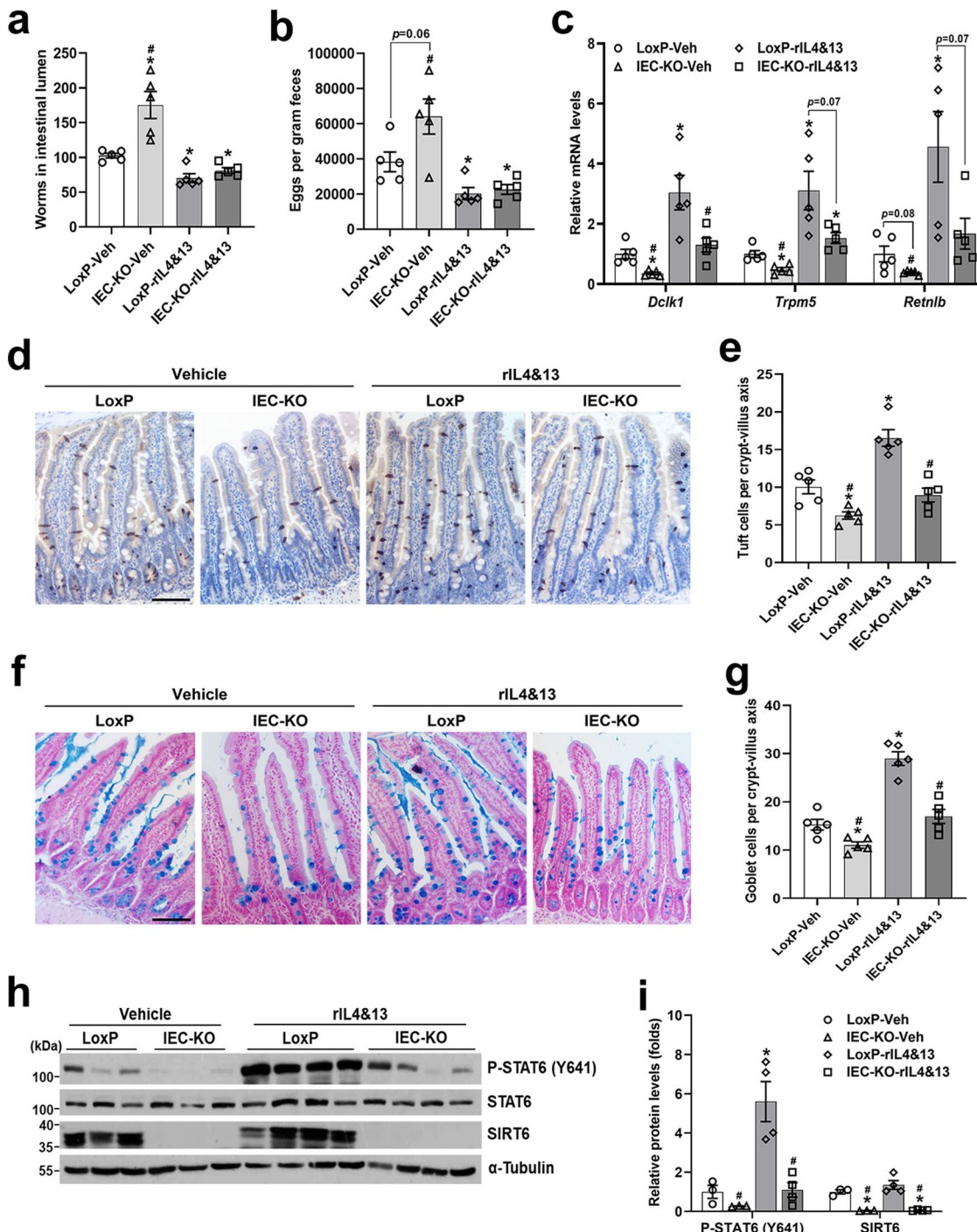

**Fig. 2 | Recombinant IL4&IL13 treatment fails to rescue the impaired tuft and goblet cell hyperplasia caused by epithelial *Sirt6* ablation.** 3-4-month-old male LoxP and IEC-KO mice were injected (*i.p.*) daily with vehicle or rIL4&rIL13 (1ug/ mouse) from day 6 after *H. poly* infection and subjected to the following analyses on day 14 post-infection. **a, b** Analysis of parasite burden by quantification of adult worms in intestinal lumen (**a**) and eggs in feces (**b**). *n* = 5 mice/group; form left to right, *$p$ = 0.019, 0.0038, 0.0038; #$p$ = 0.004 in (**a**); *$p$ = 0.03, 0.0466; #$p$ = 0.0092 in (**b**). **c** qPCR analysis of tuft and goblet cell markers expression in the jejunal IECs. *n* = 5 mice/group; form left to right, *Dclk1* (*$p$ = 0.0125, 0.021; #$p$ = 0.0092, 0.0348), *Trpm5* (*$p$ = 0.0058, 0.0288, 0.0422; #$p$ = 0.0144), *Retnlb* (*$p$ = 0.0368; #$p$ = 0.0235). **d** Jejunal tuft cells were examined by IHC staining with anti-DCLK1 antibody (200X). **e** Quantification of tuft (DCLK1 positive) cells shown in (**d**). *n* = 5

mice/group; 20 crypt-villus units counted for each mouse; from left to right, *$p$ = 0.0104, 0.0021; #$p$ = 0.0002, 0.0008. **f** Jejunal goblet cells were examined by Alcian blue staining (200X). **g** Quantification of goblet (Alcian blue positive) cells shown in (**f**). *n* = 5 mice/group; 20 crypt-villus units counted for each mouse; from left to right, *$p$ = 0.0149, <0.0001; #$p$ < 0.0001, 0.0003. **h, i** Western blot (**h**) and quantification (**i**) analyses of P-STAT6 (Y641) and SIRT6 in the jejunal IECs. n = 3, 3, 4, 4 mice/group, respectively; from left to right, P-STAT6 (*$p$ = 0.0159; #$p$ = 0.0136, 0.0161), SIRT6 (*$p$ = 0.0118, 0.0113; #$p$ = 0.0099, 0.0102). Data are presented as mean ± SEM. All $p$ values were generated by two-tailed unpaired $t$ test. *$p$ < 0.05 vs LoxP-Veh, #$p$ < 0.05 vs LoxP-rIL4&13. Scale bars, 100 μm. Source data are provided as a Source Data file.

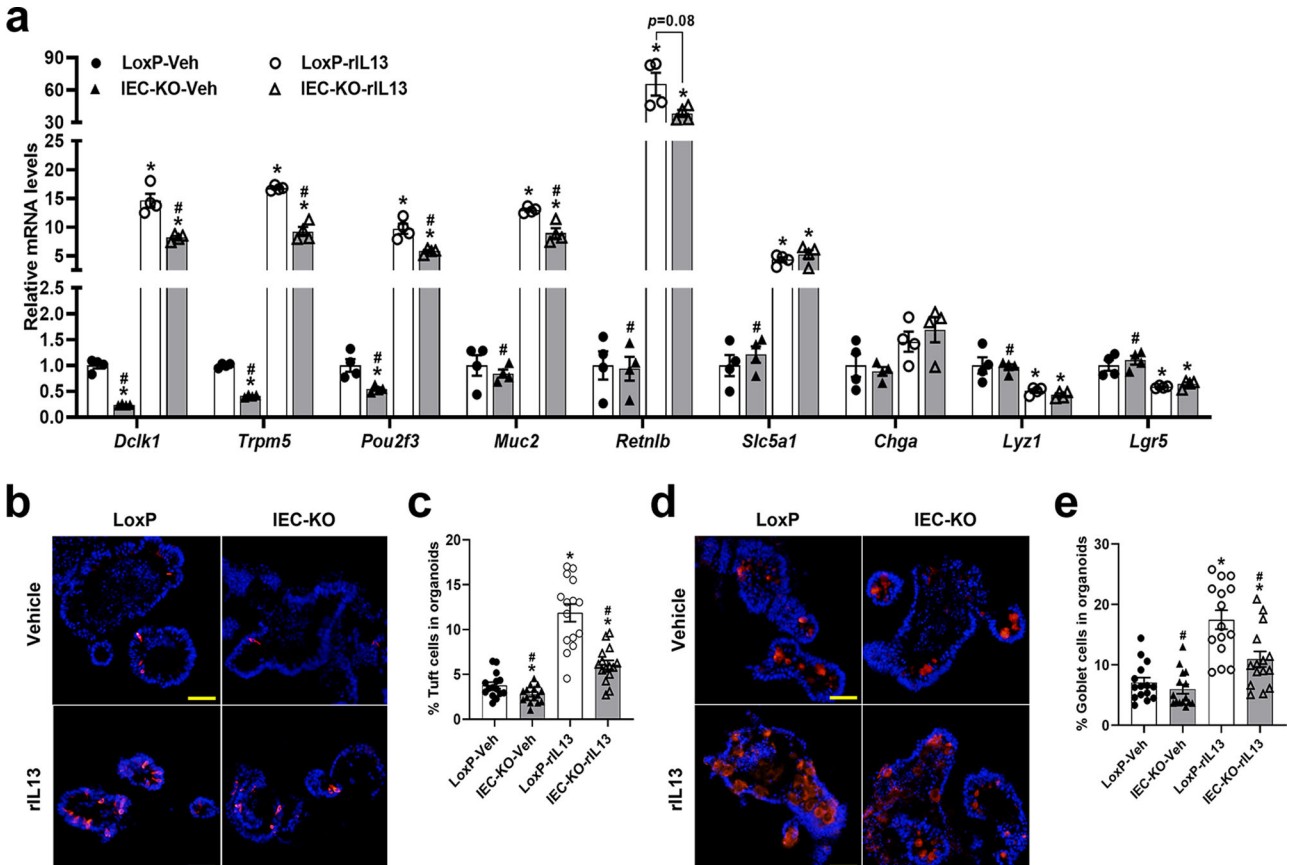

**Fig. 3 | Loss of SIRT6 in intestinal organoids causes defective IL13-induced tuft and goblet cell hyperplasia.** LoxP and IEC-KO intestinal organoids were treated with vehicle or rIL13 (25 ng/ml) for 48 h and then subjected to the following analyses. **a** qPCR analysis of IEC markers expression. $n = 4$ biological replicates/group; from left to right, *Dclk1* (*$p = 0.0001$, 0.0014, 0.0001; #$p = 0.0012$, 0.01); *Trpm5* (*$p < 0.0001$, <0.0001, 0.0018; #$p < 0.0001$, 0.0015); *Pou2f3* (*$p = 0.0303$, 0.0018, <0.0001; #$p = 0.0018$, 0.017); *Muc2* (*$p < 0.0001$, 0.002; #$p < 0.0001$, 0.0154); *Retnlb* (*$p = 0.0087$, 0.0013; #$p = 0.0087$); *Slc5a1* (*$p = 0.0019$, 0.0116; #$p = 0.0032$); *Lyz1* (*$p = 0.049$, 0.0339; #$p = 0.0008$); *Lgr5* (*$p = 0.0265$, 0.0326; #$p = 0.0082$).

**b, d** Tuft and goblet cells were labeled by anti-DCLK1 (**b**) and anti-MUC2 (**d**), respectively, in frozen sections of organoids (200X). **c** Quantification of tuft cells shown in (**b**). $n = 15$ organoids/group; from left to right, *$p = 0.0301$, <0.0001, 0.001; #$p < 0.0001$ for both. **e** Quantification of goblet cells shown in (**d**). $n = 15$ organoids/group; from left to right, *$p < 0.0001$, 0.0176; #$p < 0.0001$, 0.0032. Data are presented as mean ± SEM. All $p$ values were generated by two-tailed unpaired $t$ test. *$p < 0.05$ vs LoxP-Veh, #$p < 0.05$ vs LoxP-rIL13. Scale bars, 50 μm. Source data are provided as a Source Data file.

organoids with DCLK1 or MUC2 antibodies. Consistently, vehicle-treated IEC-KO organoids had fewer tuft cells than LoxP organoids. As expected, treatment of IEC-KO organoids with rIL13 triggered significantly less tuft cell hyperplasia than that detected in LoxP organoids (Fig. 3b, c). Although the goblet cell numbers were comparable in vehicle-treated LoxP and IEC-KO organoids, we observed a significant reduction of IL13-induced goblet cell expansion in IEC-KO organoids compared with LoxP organoids (Fig. 3d, e).

**Epithelial deletion of *Sirt6* causes defective STAT6 activity**

To uncover the functional link between SIRT6 and IL4/IL13-STAT6 pathway, we first analyzed the expression of SIRT6 and phosph-STAT6 (Y641) in the jejunal epithelium of mice infected with *H. poly* for different time intervals. Immunostaining analysis demonstrated that SIRT6 and P-STAT6 (Y641) were both primarily localized in the nucleus of crypt epithelial cells (Fig. 4a, b). Importantly, along with the rapid induction of STAT6 (Y641) phosphorylation, *H. poly* infection noticeably increased and sustained SIRT6 expression (Fig. 4a, b). It is worthy to note that the specificity of SIRT6 and P-STAT6 (Y641) antibodies was validated by immunostaining of the jejunum tissues from IEC-KO and *Stat6*[-/-] mice, respectively (Supplementary Fig. 5a, b). Consistently, western blot results revealed that the protein levels of SIRT6 and P-STAT6 (Y641) were markedly increased on day 6 p.i. and kept high

levels to day 14 p.i., indicative of the potential involvement of SIRT6 in regulating anti-helminthic STAT6 activity (Fig. 4c, d). We went on to examine whether *Sirt6* deletion leads to impaired STAT6 phosphorylation. As expected, STAT6 (Y641) phosphorylation was significantly reduced in the jejunal IECs of IEC-KO mice in both naive and *H. poly*-infected states (Fig. 4e–g). In addition, despite the absence of detectable STAT6 (Y641) phosphorylation in vehicle-treated intestinal organoids (probably due to the lack of type 2 cytokines in the organoid culture system), IEC-KO organoids had significantly reduced IL13-induced STAT6 (Y641) phosphorylation compared with LoxP organoids (Fig. 4h, i).

Furthermore, we investigated the effects of SIRT6 on the transcriptional activity of STAT6 using luciferase reporter assay in HEK293T cells. It should be pointed out that HEK293T cells lack endogenous STAT6, but they express all other components of the IL4/IL13 signaling and are able to activate an IL4/IL13 responsive promoter upon ectopic expression of STAT6[34]. Using a luciferase reporter containing two tandem copies of STAT6 binding sites, we found that SIRT6 overexpression promoted, while SIRT6 knockdown suppressed the transcriptional activity of STAT6 in the absence or presence of rIL13 stimulation (Fig. 8t, u). Together, these results suggest that SIRT6 is able to augment STAT6 activity by increasing IL13-induced Tyr641 phosphorylation.

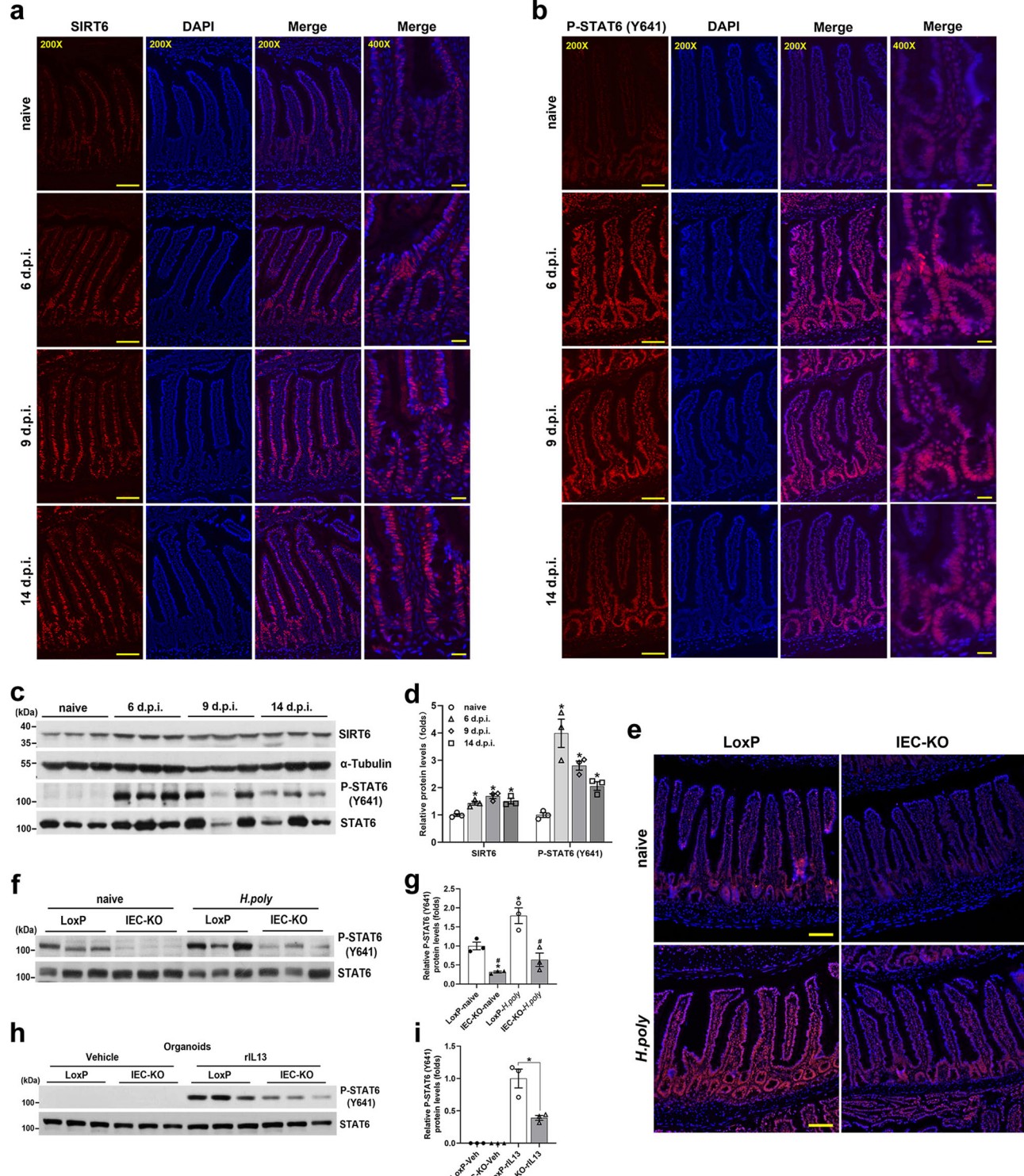

**Fig. 4 | IEC ablation of *Sirt6* attenuates the Tyr641 phosphorylation of STAT6.** **a**–**d** 2-month-old C57BL/6J male mice were infected with *H. poly* and analyzed on indicated days post-infection. **a**, **b** Immunostaining for SIRT6 (**a**) and P-STAT6 (Y641) (**b**) in the jejunum. **c**, **d** Western blot (**c**) and quantification (**d**) analyses of SIRT6 and P-STAT6 (Y641) in the jejunal IECs. $n = 3$ mice/group; $p = 0.0102$ (6 dpi), 0.006 (9 dpi), 0.0203 (14 dpi) for SIRT6; $p = 0.0259$ (6 dpi), 0.0025 (9 dpi), 0.0089 (14 dpi) for P-STAT6. **e**–**g** 2–3-month-old male naive and *H. poly*-infected LoxP and IEC-KO mice were subjected to the following assays on day 14 post-infection.

**e** Immunostaining for P-STAT6 (Y641) in the jejunum. **f**, **g** Western blot (**f**) and quantification (**g**) analyses of P-STAT6 (Y641) in the jejunal IECs. $n = 3$ mice/group; from left to right, $*p = 0.0166, 0.0423$; $\#p = 0.0177, 0.0136$. **h**, **i** Western blot (**h**) and quantification (**i**) analyses of P-STAT6 (Y641) in vehicle- or rIL13-treated (25 ng/ml, 48 h) organoids. $n = 3$ biological replicates/group; $p = 0.0451$. Data are presented as mean ± SEM. All $p$ values were generated by two-tailed unpaired $t$ test. In **d**, $*p < 0.05$ vs naive; In **g**, $*p < 0.05$ vs LoxP-naive, $\#p < 0.05$ vs LoxP-*H. poly*; Scale bars, 200X, 100 μm; 400X, 25 μm. Source data are provided as a Source Data file.

## Transgenic mice with IEC-specific overexpression of constitutively activated STAT6 exhibits enhanced tuft and goblet cell differentiation

To directly explore the role of STAT6 in intestinal epithelium, we generated transgenic mice that express a constitutively active form of mouse STAT6 (STAT6vt), driven by mouse *Vil1* gene promoter (TgSTAT6vt) (Fig. 5a). STAT6vt contains two amino acid mutations (V547A/T548A) in the SH2 domain resulting in Jak kinase independent Tyr641 phosphorylation[35]. We obtained 3 transgenic mouse lines (line-10, line-13 and line-18) expressed transgene STAT6vt at different levels. qPCR analysis of the jejunal IECs identified line-18 as the highest expresser of STAT6vt (~22-fold increase), followed by line-13 (~7-fold increase) and line-10 (~2-fold increase) (Fig. 5c). The protein levels of STAT6vt in these 3 transgenic lines were also confirmed by western blot analysis (Fig. 5b). Importantly, STAT6vt was only expressed in the intestine but not in other tissues including stomach, liver, lung, heart,

and kidney of line-18 mice, indicating an intestinal epithelial-specific expression pattern of STAT6vt (Supplementary Fig. 6a). As we showed above, line-13 transgenic mice exhibited moderate epithelial expression of STAT6vt and no outwardly apparent phenotype, we therefore selected transgenic line-13 for further study (hereafter referred to as TgSTAT6vt).

Intriguingly, TgSTAT6vt mice had reduced jejunal villus height and crypt depth compared with WT controls (Fig. 5d, e). Furthermore, qPCR analysis revealed a significantly augmented expression of genes encoding tuft (*Dclk1*, *Trpm5*) and goblet (*Retnlb*) markers in TgSTAT6vt mice compared with WT mice, indicating STAT6 in IEC promotes tuft and goblet cell differentiation (Fig. 5f). In contrast, we observed a significant reduction of *Lyz1* expression in the jejunal IECs of TgSTAT6vt mice (Fig. 5f). No changes in the expression of genes encoding *Slc5a1* (enterocyte), *Chga* (enteroendocrine cell), and *Lgr5* (stem cell) were detected (Fig. 5f). To exclude the possibility that

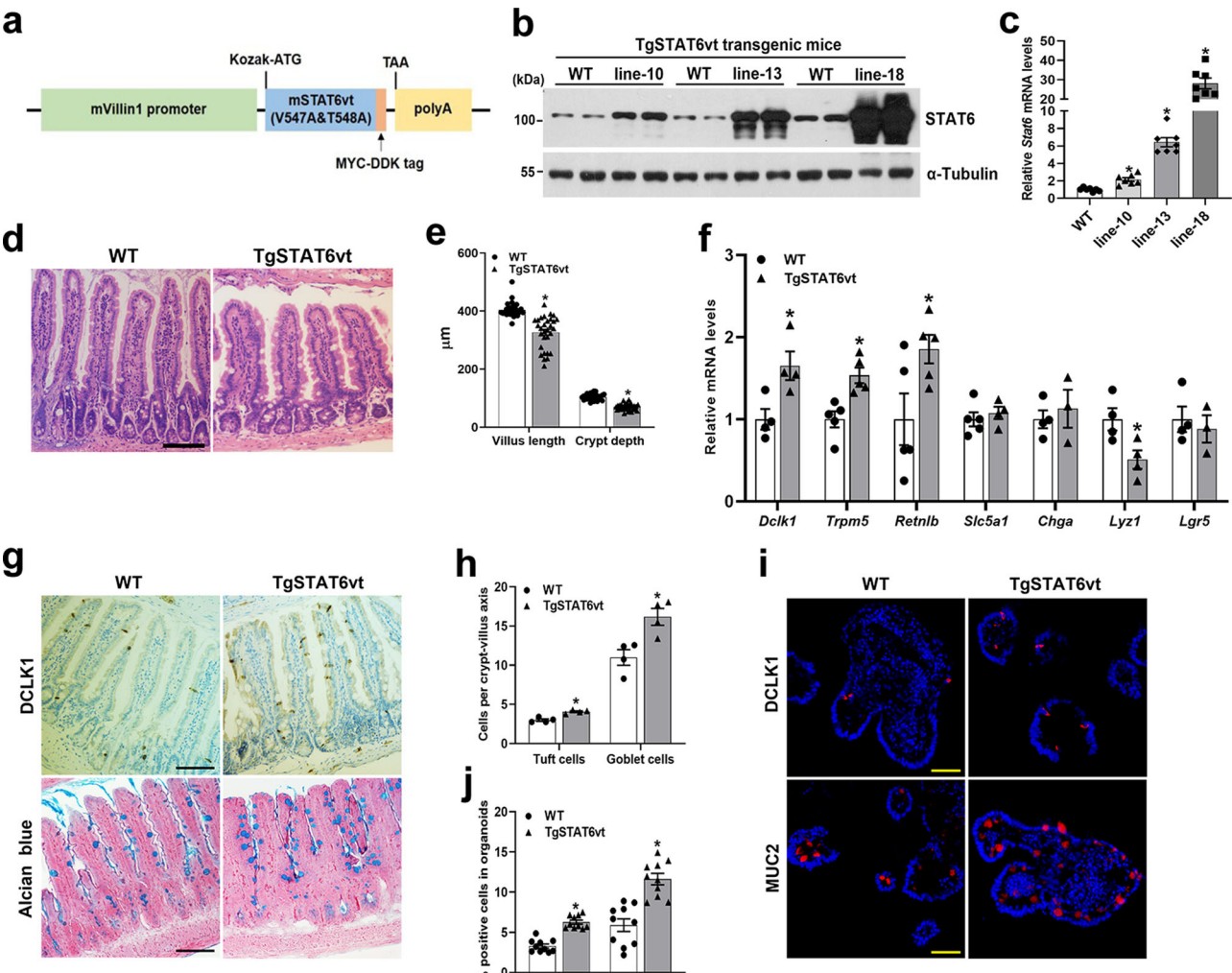

**Fig. 5 | IEC-specific overexpression of constitutively activated STAT6 (STAT6vt) promotes tuft and goblet cell differentiation.** 2-month-old male WT and TgSTAT6vt mice (including line-10, line-13 and line-18) were subjected to the following assays. **a** The strategy for creating TgSTAT6vt mice. **b, c** Western blot (**b**) and qPCR analyses (**c**) of TgSTAT6vt expression in the jejunal IECs. $n = 7$ mice/group in (**c**); $p = 0.0013$ (line-10), <0.0001 (line-13), <0.0001 (line-18). **d** Representative H&E images of jejunum tissues. **e** Analyses of jejunal villus length and crypt depth. $n = 30$ villi or crypts/group; $p < 0.0001$ (villus length, crypt depth). **f** qPCR analysis of IEC markers expression in the jejunal IECs. *Dclk1*, *Lyz1*, $n = 4$ mice/group; *Trpm5*, *Retnlb*, $n = 5$ mice/group; *Slc5a1*, $n = 5$ (WT) and 4 (Tg) mice; *Chga*, *Lgr5*, $n = 4$ (WT) and 3 (Tg) mice; $p = 0.0259$ (*Dclk1*), 0.0044 (*Trpm5*), 0.0491

(*Retnlb*), 0.0322 (*Lyz1*). **g** Tuft and goblet cells were examined by DCLK1 immunostaining and Alcian blue staining, respectively in the jejunum (200X). **(h)** Quantification of tuft and goblet cells shown in (**g**). $n = 4$ mice/group; 20 crypt-villus units counted for each mouse; $p = 0.0018$ (tuft cell) and 0.0126 (goblet cell). **i** Tuft and goblet cells were labeled by anti-DCLK1 and anti-MUC2, respectively, in frozen sections of organoids (200X). **j** Quantification of tuft and goblet cells shown in (**i**). $n = 10$ organoids/group; $p < 0.0001$ (tuft cell, goblet cell). Data are presented as mean ± SEM. All $p$ values were generated by two-tailed unpaired $t$ test. *$p < 0.05$. Scale bars, 100 μm in (**d, g**); 50 μm in (**i**). Source data are provided as a Source Data file.

intestinal epithelial phenotype observed in TgSTAT6vt mice was caused by the side effects of transgene random insertion, we also examined the mRNA expression of IEC lineage markers in the jejunal IECs of line-18 mice. Consistently, mRNA levels of *Dclk1* and *Retnlb* in line-18 mice were significantly higher than those in WT mice, while *Lyz1* mRNA expression was dramatically lower (Supplementary Fig. 6d). Interestingly, while the epithelial expression of *Slc5a1*, *Chga*, and *Lgr5* were comparable in WT and TgSTAT6vt (line-13) mice, their levels were significantly lower in line-18 mice than in control mice (Fig. 5f, Supplementary Fig. 6d). This discrepancy is likely attributed to the different expression levels of transgene STAT6vt between these 2 mouse lines. Notably, we also observed a more dramatic reduction of jejunal villus height in line-18 mice, indicating epithelial STAT6 directly regulates villus development in the SI (Supplementary Fig. 6b, c).

To further analyze whether augmented expression of cell markers correlated with the increased tuft and goblet cell numbers in TgSTAT6vt mice, we stained jejunum tissues for DCLK1 and Alcian blue. As expected, TgSTAT6vt mice had more tuft and goblet cells in the jejunal epithelium than WT mice (Fig. 5g, h). To test if tuft and goblet cell expansion is an IEC-intrinsic regulation by STAT6, we examined the tuft and goblet cell abundance in the intestinal organoids. Indeed, TgSTAT6vt organoids had significantly more tuft and goblet cells than WT organoids, revealing an epithelium-autonomous effect of STAT6 in tuft and goblet cell differentiation (Fig. 5i, j).

### IEC STAT6vt overexpression rescues impaired intestinal anti-helminth responses in IEC-KO mice

To better understand the transcriptional program regulated by SIRT6 in the intestinal epithelium, we conducted RNA-seq analysis on the jejunal IECs from LoxP and IEC-KO mice. Among the 188 significantly differentially expressed genes (log2(fold-change)>1.25, and *p* < 0.05), 58 genes were downregulated and 130 genes were upregulated (Supplementary Fig. 7a). Gene Set Enrichment Analysis (GSEA) revealed that both tuft-1 and tuft-2 identity gene sets were enriched in downregulated genes by *Sirt6* deficiency, again supporting that epithelial SIRT6 is critical for intestinal tuft cell development (Supplementary Fig. 7b)[36]. Notably, unbiased gene ontology analysis revealed that pathways like positive regulation of inflammatory response, anatomical structure homeostasis and biosynthesis of specialized pro-resolving mediators (SPMs) were downregulated while pathways involved in leukocyte activation, regulation of defense response, and regulation of leukocyte cell-cell adhesion were upregulated (Supplementary Fig. 7c). Furthermore, to gain insight into the underlying molecular mechanisms of STAT6 regulated IEC development, we also performed RNA-seq analysis of the jejunal IECs from TgSTAT6vt mice and WT littermates (Supplementary Fig. 7d). When compared the gene expression changes (log2(fold-change) from DESeq2) of IEC-KO (normalized to LoxP) and TgSTAT6vt (normalized to WT), we found that many TgSTAT6vt downregulated genes were upregulated in *Sirt6* IEC-KO mice (Supplementary Fig. 7e), indicating that STAT6 and SIRT6 may be involved in the same pathway to regulate intestinal epithelial homeostasis.

Since *Sirt6* deletion leads to the compromised activation of STAT6 in response to type 2 cytokines in IECs, we hypothesized that overexpression of constitutively activated STAT6vt may rescue the epithelial abnormalities in IEC-KO mice. To test this hypothesis, we crossed IEC-KO mice with TgSTAT6vt mice to generate IEC-KO-TgSTAT6vt mice. Indeed, IEC-KO-TgSTAT6vt mice had significantly more tuft and goblet cells in the jejunal epithelium than IEC-KO mice (Supplementary Fig. 8a, b). This rescue effect was also reflected by the increased expression of *Dclk1*, *Trpm5*, and *Retnlb* in the IECs of IEC-KO-TgSTAT6vt mice (Supplementary Fig. 8c). Furthermore, a significant increase in the numbers of tuft and goblet cells were observed in IEC-KO-TgSTAT6vt organoids (Supplementary Fig. 8d, e).

To further determine whether the impaired anti-helminth responses in IEC-KO mice were due to defective IEC STAT6 activity,

we infected WT, TgSTAT6vt, IEC-KO, IEC-KO-TgSTAT6vt mice with *H. poly* and analyzed epithelial responses and worm burden in these mice on day 14 after infection. The overexpression of STAT6vt and the ablation of *Sirt6* in jejunal epithelium were confirmed by western blot analysis (Fig. 6a, b). Following infection, TgSTAT6vt mice had obviously more jejunal tuft and goblet cells than WT mice (Fig. 6d-f). Intriguingly, although the IEC-KO mice had the lowest abundance of tuft and goblet cells among the 4 groups of mice we studied, we observed comparable numbers of jejunal tuft and goblet cells in TgSTAT6vt and IEC-KO-TgSTAT6vt mice (Fig. 6d-f). These findings were also verified by qPCR analysis of tuft (*Dclk1*, *Trpm5*) and goblet (*Retnlb*) cell markers (Fig. 6c). In accord with a critical role of tuft and goblet cell expansion in *H. poly* worm expulsion, we found that IEC-KO mice had markedly higher worm burden than WT mice, while IEC overexpression of STAT6vt significantly relieved the worm burden in both WT and IEC-KO mice, as shown by fewer adult worms in the SI and eggs in the feces (Fig. 6g, h). In addition, organoids derived from these 4 groups of mice were cultured and stimulated with rIL13. Consistently, we observed comparable numbers of tuft and goblet cells in TgSTAT6vt and IEC-KO-TgSTAT6vt organoids (Fig. 6i–k). Together, these results reveal that overexpression of STAT6vt in IECs remarkably rescued the impaired intestinal type 2 immunity caused by *Sirt6* deletion.

### IEC overexpression of SIRT6 in mice promotes intestinal type 2 immune responses

To further confirm that SIRT6 indeed modulates type 2 immunity-induced intestinal epithelial remodeling, SIRT6 IEC-specific transgenic mice (IEC-Tg) were generated by crossing transgene *SIRT6* floxed mice which encompass a floxed STOP cassette in front of the *SIRT6* transgene, with *Villin-Cre* mice[37,38]. The SIRT6 protein was increased by 4- to 5-fold in the IEC-Tg jejunal epithelium (Fig. 7a, b). Intriguingly, naive WT and IEC-Tg mice had comparable abundance of tuft and goblet cells in the jejunum (Supplementary Fig. 9a–c).

WT and IEC-Tg mice were then orally inoculated with *H. poly* and analyzed intestinal epithelial responses on day 14 post-infection. In contrast to what we observed in IEC-KO mice, both tuft and goblet cell numbers and their marker genes expression were significantly elevated in IEC-Tg mice compared with WT littermates (Fig. 7c–e). Notably, IEC-Tg mice had elevated mRNA levels of type 2 cytokines and IL13 protein levels in the jejunum, as illustrated by qPCR and ELISA results, respectively (Fig. 7f, g). As a result, we observed fewer luminal worms and fecal eggs in IEC-Tg mice, indicating IEC-Tg mice are more resistant to *H. poly* infection than WT mice (Fig. 7h, i). To further demonstrate a role for SIRT6 in promoting type 2 immunity-induced tuft and goblet cell differentiation, we set up intestinal organoid cultures from WT and IEC-Tg mice and treated them with rIL13. Expectedly, we observed that IEC-Tg organoids had more tuft and goblet cells than WT controls after rIL13 treatment (Fig. 7j–l). Importantly, P-STAT6 (Y641) levels in the jejunal IECs were significantly elevated in *H. poly*-infected IEC-Tg mice compared with WT littermates, indicating that SIRT6 is sufficient to enhance STAT6 activity when type 2 immunity is activated by *H. poly* infection (Fig. 7a, m). Consistently, we also observed substantially increased P-STAT6 (Y641) levels in rIL13-treated organoids of IEC-Tg mice (Fig. 7n, o). Collectively, these data show that epithelial SIRT6 is sufficient to drive helminth infection-induced epithelial remodeling through modulating STAT6 activity.

### SIRT6 upregulates the phosphorylation of STAT6 (Tyr641) through suppressing SOCS3 expression

Next, we sought to determine the underlying molecular mechanisms of how SIRT6 activates STAT6. Since acetylation of STAT6 is required for its transcriptional activity[39,40], it is unlikely that SIRT6, a NAD+-dependent protein deacetylase, activates STAT6 through deacetylation. Indeed, overexpression of SIRT6 did not affect the acetylation of STAT6

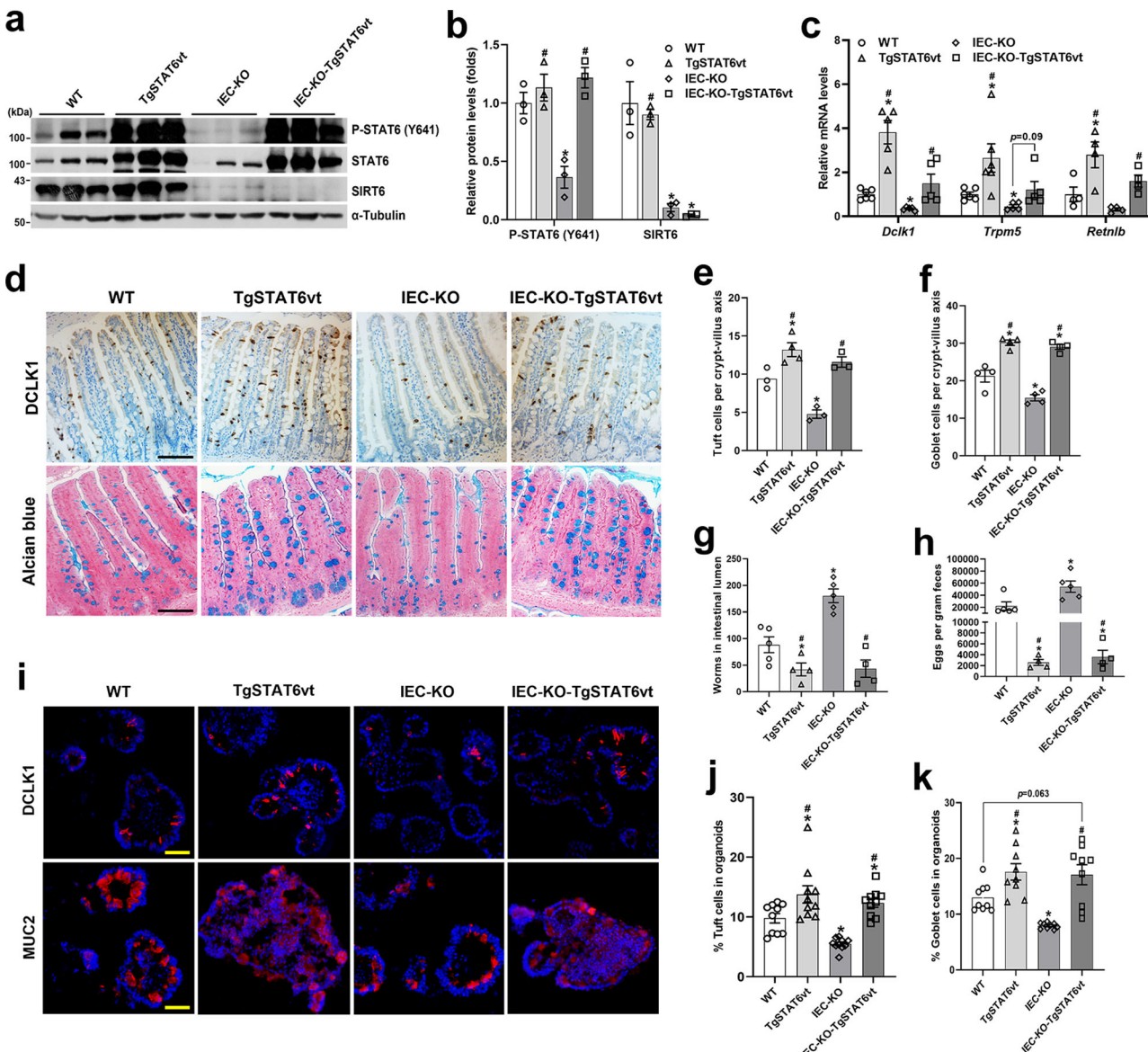

**Fig. 6 | IEC STAT6vt overexpression rescues the defective intestinal type 2 immunity resulting from *Sirt6* deletion.** 2–3-month-old male mice with 4 genotypes (WT, TgSTAT6vt, IEC-KO, IEC-KO-TgSTAT6vt) were infected with *H. poly* and subjected to the following analyses on day 14 post-infection. **a, b** Western blot (**a**) and quantification (**b**) analyses of P-STAT6 (Y641) and SIRT6 in the jejunal IECs. *n* = 3 mice/group; from left to right, P-STAT6 (**\****p* = 0.0083; #*p* = 0.0074, 0.0026); SIRT6 (**\****p* = 0.0358, 0.0344; #*p* = 0.0002). **c** qPCR analysis of the expression of *Dclk1*, *Trpm5* and *Retnlb* in the jejunal IECs. *Dclk1*, *Trpm5*, *n* = 5 mice/group; *Retnlb*, *n* = 4 mice/group; from left to right, *Dclk1* (**\****p* = 0.0054, 0.0019; #*p* = 0.0027, 0.0494); *Trpm5* (**\****p* = 0.0389, 0.0085; #*p* = 0.0184); *Retnlb* (**\****p* = 0.0466; #*p* = 0.0232, 0.0154). **d** Tuft and goblet cells were examined by DCLK1 immunostaining and Alcian blue staining, respectively in the jejunum (200X). **e, f** Quantification of tuft (**e**) and goblet (**f**) cells shown in (**d**). *n* = 3, 4, 3, 3 mice/ group, respectively in (**e**) and *n* = 4 mice/group in (**f**); 30 crypt-villus units counted for each mouse; form left to right, **\****p* = 0.0253, 0.0119; #*p* = 0.0007, 0.0017 in (**e**); **\****p* = 0.0059, 0.0264, 0.0104; #*p* < 0.0001 for both in (**f**). **g, h** Analysis of parasite burden by quantification of adult worms in intestinal lumen (**g**) and eggs in feces (**h**). *n* = 5, 4, 5, 4 mice/group; form left to right, **\****p* = 0.0458, 0.0016; #*p* = 0.0001, 0.0005 in (**g**); **\****p* = 0.0487, 0.0268; #*p* = 0.0051, 0.0052 in (**h**). **i** Tuft and goblet cells were labeled by anti-DCLK1 and anti-MUC2, respectively, in frozen sections of rIL13-treated organoids (200X). **j, k** Quantification of tuft (**j**) and goblet (**k**) cells shown in (**i**). *n* = 10 (**j**) and 9 (**k**) organoids/group; form left to right, **\****p* = 0.0307, 0.0003, 0.0281; #*p* = 0.0003, <0.0001 in (**j**); **\****p* = 0.0181, 0.0002; #*p* = 0.0002, 0.0009 in (**k**). Data are presented as mean ± SEM. All *p* values were generated by two-tailed unpaired *t* test. **\****p* < 0.05 vs WT, #*p* < 0.05 vs IEC-KO. Scale bars, 100 μm in (**d**); 50 μm in (**i**). Source data are provided as a Source Data file.

in either HEK293T cells or in NCM460 cells (a normal human colon mucosal epithelial cell line), indicating that STAT6 is not a substrate for SIRT6 deacetylase activity (Supplementary Fig. 10a, b). Considering that SIRT6 can repress gene expression through histone deacetylation-related chromatin remodeling, it is reasonable to postulate that *Sirt6* deletion in IECs leads to upregulated expression of IL4/IL13-STAT6 signaling inhibitors, thereby attenuating the activity of STAT6. Suppressor of cytokine signaling 3 (SOCS3), a well-studied suppressor of JAK/STAT signaling[41], has attracted our attention because its

expression was negatively associated with SIRT6 expression in the jejunal IECs of mice infected with *H. poly* (Fig. 4c, d and 8a–c). More importantly, we found that the expression of SOCS3 in the jejunal epithelium was upregulated in *H. poly*-infected IEC-KO mice whereas downregulated in *H. poly*-infected IEC-Tg mice as compared with their corresponding control mice (Fig. 8d–i). These in vivo results were further confirmed in rIL13-treated NCM460 cells with SIRT6 knockdown or overexpression. Accordingly, knockdown of SIRT6 significantly upregulated the expression of SOCS3 but attenuated IL13-induced

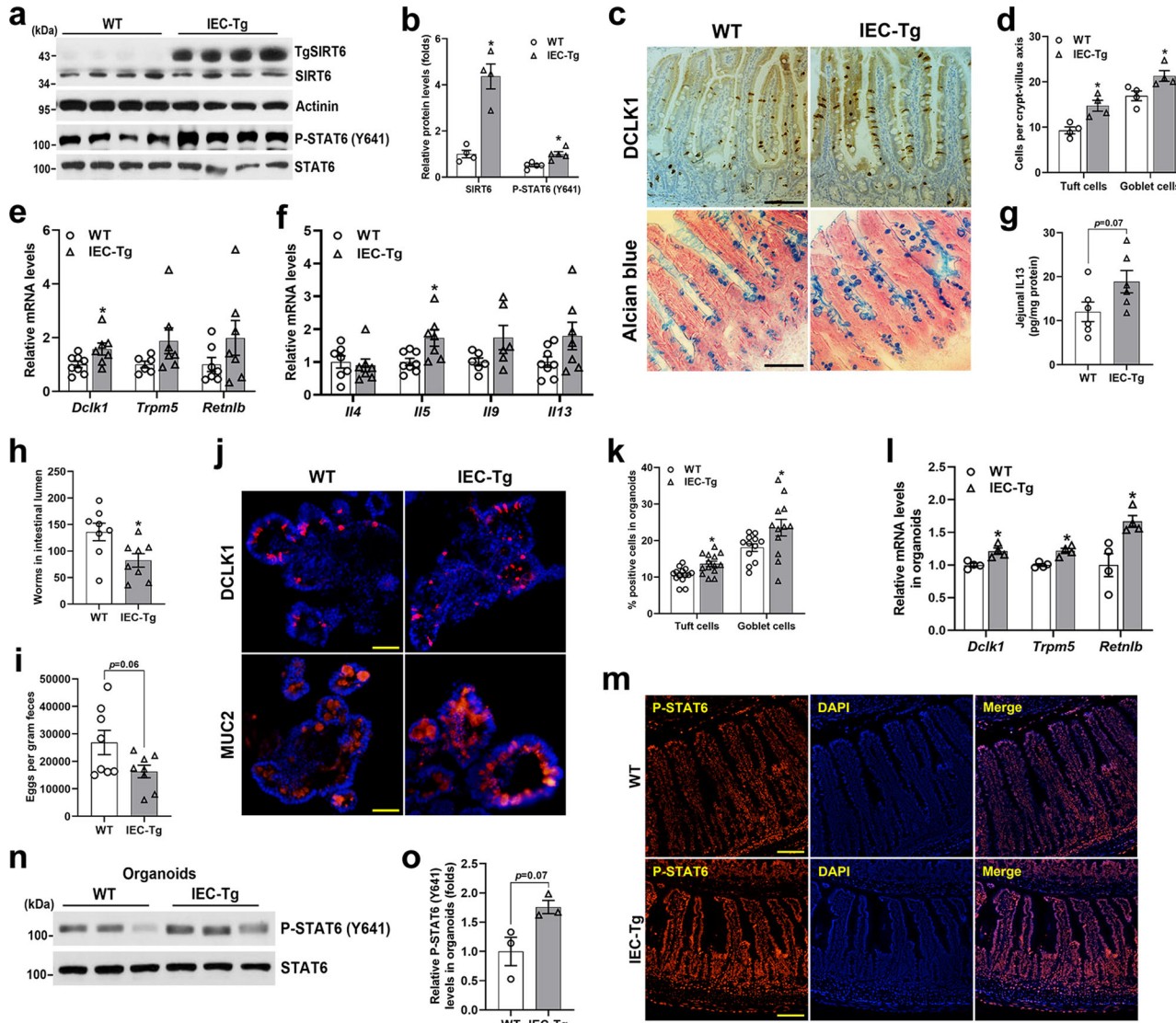

**Fig. 7 | IEC overexpression of SIRT6 elevates the Tyr641 phosphorylation of STAT6 and enhances anti-helminth responses.** 3-month-old male WT and IEC-Tg mice were infected with *H. poly* and subjected to the following analyses on day 14 post-infection. **a, b** Western blot (**a**) and quantification (**b**) analyses of SIRT6 and P-STAT6 (Y641) in the jejunal IECs. n = 4 mice/group (SIRT6) and 5 mice/group (P-STAT6); p = 0.0057 (SIRT6, P-STAT6). **c** Tuft and goblet cells were examined by DCLK1 immunostaining and Alcian blue staining, respectively, in the jejunum (200X). **d** Quantification of tuft and goblet cells shown in (**c**). n = 4 mice/group; 20 crypt-villus units counted for each mouse; p = 0.0118 (tuft cell) and 0.0338 (goblet cell). **e** qPCR analysis of tuft and goblet cell markers expression in the jejunal IECs. *Dclk1*, n = 8 (WT) and 7 (IEC-Tg) mice; *Trpm5*, n = 6 (WT) and 7 (IEC-Tg) mice; *Retnlb*, n = 7 mice/group; p = 0.0489 (*Dclk1*). **f** qPCR analysis of type 2 immune cytokine genes expression in the jejunum. *Il4*, n = 7 mice/group; *Il5, Il13*, n = 8 (WT) and 7 (IEC-Tg) mice; *Il9*, n = 6 mice/group; p = 0.028 (*Il5*). **g** Jejunal IL13 levels were

measured by ELISA. n = 6 mice/group. **h, i** Analysis of parasite burden by quantification of adult worms in intestinal lumen (**h**) and eggs in feces (**i**). n = 8 mice/group; p = 0.0227 in (**h**). **j** Tuft and goblet cells were labeled by anti-DCLK1 and anti-MUC2, respectively, in frozen sections of rIL13-treated (25 ng/ml, 48 h) organoids (200X). **k** Quantification of tuft and goblet cells shown in (**j**). n = 14 (tuft cell) and 12 (goblet cell) organoids/group; p = 0.0029 (tuft cell) and 0.0451 (goblet cell). **l** qPCR analysis of the expression of *Dclk1, Trpm5* and *Retnlb* in organoids treated with rIL13. n = 4 biological replicates/group; p = 0.0145 (*Dclk1*), 0.0057 (*Trpm5*), 0.0237 (*Retnlb*). **m** Immunostaining for P-STAT6 (Y641) in the jejunum. **n, o** Western blot (**n**) and quantification (**o**) analyses of P-STAT6 (Y641) in rIL13 treated organoids. n = 3 biological replicates/group. Data are presented as mean ± SEM. All p values were generated by two-tailed unpaired t test. *p < 0.05. Scale bars, 100 μm in (**c, m**); 50 μm in (**j**). Source data are provided as a Source Data file.

phosphorylation of STAT6 (Y641). Conversely, overexpression of SIRT6 lowered the SOCS3 levels while enhanced STAT6 phosphorylation (Fig. 8p–s).

To further investigate whether SIRT6 directly represses SOCS3 transcription through its histone deacetylation activity, chromatin immunoprecipitation (ChIP) was performed in rIL13-treated NCM460 cells to study whether SIRT6 binds to the promoter region of *SOCS3*. Indeed, SIRT6 was significantly enriched at the *SOCS3* gene promoter, especially in regions 2 and 3 (Fig. 8j, k). Moreover, we went on to determine whether SIRT6 inhibits *SOCS3* gene transcription via

histone deacetylation. Expectedly, we found that SIRT6 overexpression reduced while SIRT6 knockdown increased the H3K56 (SIRT6 substrate) acetylation levels in regions 2 and 3, indicating that SIRT6 suppresses *SOCS3* expression through deacetylating H3K56 (Fig. 8l). We next examined the effects of overexpression of SOCS3 on IL13-stimulated STAT6 (Y641) phosphorylation in HEK293T cells or NCM460 cells. SOCS3 overexpression remarkably reduced the IL13-induced Tyr641 phosphorylation of either ectopically-expressed STAT6 in HEK293T cells or endogenous STAT6 in NCM460 cells (Fig. 8m–o).

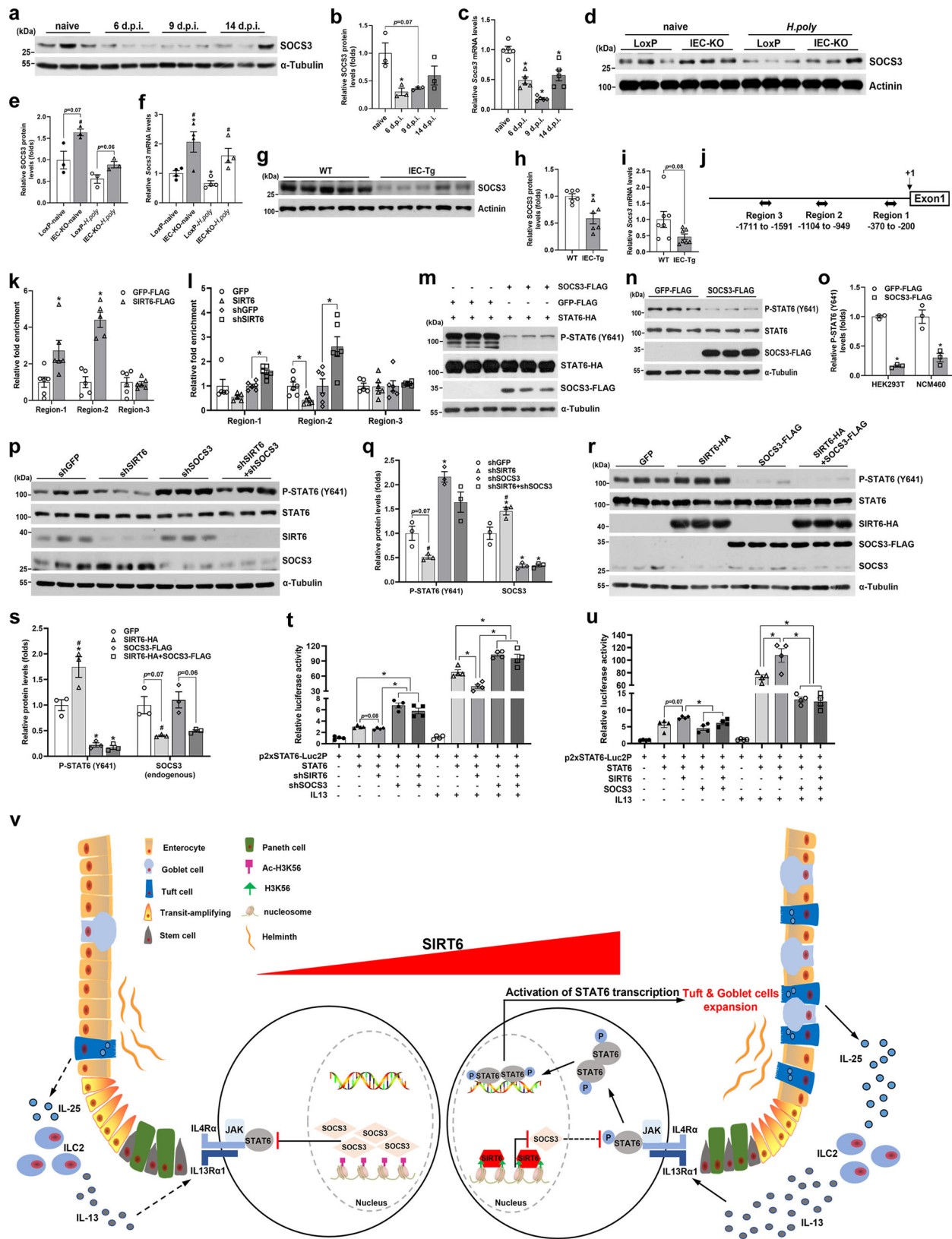

To assess whether SIRT6 is dependent on SOCS3 to regulate STAT6 phosphorylation, we knocked down (KD) or overexpressed (OE) SIRT6 and/or SOCS3 in NCM460 cells and treated cells with rIL13 to stimulate STAT6 (Y641) phosphorylation. As expected, SIRT6 KD reduced the phosphorylation of STAT6. However, SOCS3 KD led to a noticeable increase of STAT6 phosphorylation, even in SIRT6 KD cells

(Fig. 8p, q). In contrast, SIRT6 OE significantly elevated STAT6 phosphorylation, but SOCS3 OE profoundly abolished the effect of SIRT6 on increasing STAT6 phosphorylation (Fig. 8r, s). To further prove whether SIRT6 modulates STAT6 transcriptional activity, luciferase assay using p2xSTAT6-Luc2P reporter was conducted in HEK293T cells. SIRT6 KD- or SIRT6 OE-induced changes of luciferase activity was

**Fig. 8 | SIRT6 promotes the Tyr641 phosphorylation of STAT6 via inhibiting SOCS3 expression. a–c** Western blot (**a**) and quantification (**b**) and qPCR (**c**) analyses of SOCS3 expression in the jejunal IECs from 2-month-old male C57BL/6J mice on indicated days post-*H. poly* infection. $n = 3$ mice/group in (**b**) and 5 mice/group in (**c**); $p = 0.0499$ (6 dpi) in (**b**); $p = 0.0002$ (6 dpi), <0.0001 (9 dpi), 0.0058 (14 dpi) in (**c**). **d–f** Western blot (**d**) and quantification (**e**) and qPCR (**f**) analyses of SOCS3 expression in the jejunal IECs from 2–3-month-old naive and *H. poly*-infected LoxP and IEC-KO mice. $n = 3$ mice/group in (**e**) and 4 mice/group in (**f**); from left to right, #$p = 0.0012$ in (**e**); *$p = 0.0493$, 0.0378; #$p = 0.0233$, 0.0241 in (**f**). **g–i** Western blot (**g**) and quantification (**h**) and qPCR (**i**) analyses of SOCS3 expression in the jejunal IECs from 3-month-old male WT and IEC-Tg mice infected with *H. poly*. $n = 6$ mice/group in (**h**) and 7 mice/group in (**i**); $p = 0.0073$ in (**h**). **j** Schematic illustration of the primer amplicons (Regions 1–3) used for scanning ChIP analysis of the proximal *SOCS3* promoter. **k, l** The enrichment of SIRT6 (**k**) and the effects of SIRT6 overexpression or knockdown on H3K56 acetylation (**l**) within the *SOCS3* proximal promoter were analyzed using ChIP in rIL13-treated (10 ng/ml, 6 h) NCM460 cells. In (**k**), $n = 6$ (Region-1&−3) and 5 (Region-2) biological replicates/group; $p = 0.0257$ (Region-1), 0.0002 (Region-2); In (**l**), $n = 6$ biological replicates/group; GFP vs SIRT6, $p = 0.0136$ (Region-2); shGFP vs shSIRT6, $p = 0.0011$ (Region-1), 0.0101 (Region-2). **m, n** The effects of SOCS3 overexpression on STAT6 (Y641)

phosphorylation were examined by western blot in rIL13-treated (10 ng/ml, 6 h) HEK293T (**m**) and NCM460 (**n**) cells. **o** Quantification analysis of P-STAT6 (Y641) levels shown in (**m**, **n**). $n = 3$ biological replicates/group; $p < 0.0001$ (HEK293T), 0.0107 (NCM460). **p–s** Western blot (**p**, **r**) and quantification (**q**, **s**) analyses of P-STAT6 (Y641), SIRT6 and SOCS3 in NCM460 cells transfected with shSIRT6 and/or shSOCS3 constructs (**p**, **q**) or SIRT6 and/or SOCS3 constructs (**r**, **s**), followed with rIL13 stimulation (10 ng/ml, 6 h). $n = 3$ biological replicates/group; In (**q**), from left to right, *$p = 0.0039$; #$p = 0.0017$ (P-STAT6); *$p = 0.0491$, 0.0263, 0.0306; #$p = 0.0011$ (SOCS3); In (**s**), from left to right, *$p = 0.0466$, 0.0101, 0.0096; #$p = 0.0141$ (P-STAT6); #$p = 0.0457$ (SOCS3). **t, u** p2xSTAT6-Luc2P luciferase reporter assays in HEK293T cells transfected with shSIRT6 and/or shSOCS3 constructs (**t**) or SIRT6 and/or SOCS3 constructs (**u**), followed with vehicle or rIL13 stimulation (10 ng/ml, 6 h). $n = 4$ biological replicates/group. In (**t**), from left to right, $p = 0.0098$, 0.0075, 0.005, 0.0343, 0.0027, respectively. In (**u**), from left to right, $p = 0.0459$, 0.0416, 0.0003, 0.003, respectively. **v** Schematic illustration of the role of IEC SIRT6 in regulation of helminth-induced epithelial remodeling. Data are presented as mean ± SEM. All $p$ values were generated by two-tailed unpaired $t$ test. *$p < 0.05$; In (**b**, **c**), *$p < 0.05$ vs naive; In (**e**, **f**), *$p < 0.05$ vs LoxP-naive, #$p < 0.05$ vs LoxP-*H. poly*; In (**q**), *$p < 0.05$ vs shGFP, #$p < 0.05$ vs shSOCS3; In (**s**), *$p < 0.05$ vs GFP, #$p < 0.05$ vs SOCS3-FLAG. Source data are provided as a Source Data file.

almost completely rescued by SOCS3 KD or SOCS3 OE, respectively, especially in the presence of rIL13 stimulation (Fig. 8t, u). Collectively, these results suggest that SIRT6 promotes STAT6 Tyr641 phosphorylation and its transcriptional activity through suppressing SOCS3 transcription.

## Discussion

In the current study, we reveal a previously unrecognized function of SIRT6 in the regulation of intestinal type 2 immune responses to protect against the infection of helminth parasites. By using IEC-specific SIRT6 knockout and transgenic mice, we demonstrate that SIRT6 is required and sufficient for helminth-induced intestinal tuft and goblet cell hyperplasia. In the intestinal epithelium, the expression of SIRT6 is not limited to specific cell types, but is enriched in the crypt epithelial cells, including ISCs and TA cells. Upon activation by the invasion of helminth, intestinal tuft cells initiate type 2 immune responses by secreting IL25. IL25 acts on ILC2 in the LP to produce IL13, which directs stem and progenitor cells in the crypt to preferentially differentiated into tuft and goblet cells[11–13]. It is reasonable to postulate that SIRT6 in the intestinal progenitor cells promotes the differentiation toward tuft and goblet cells in response to helminth infection. Mature tuft cells are known as important sentinels in the gastrointestinal tract, relying on chemosensory membrane receptors to detect the presence of helminths and in turn secrete effector molecules, IL-25 and Cysteinyl leukotrienes[42,43]. Since SIRT6 is also expressed by mature tuft cells, whether SIRT6 is necessary for maintaining physiological function of mature tuft cells requires future studies.

The key question is the mechanism through which IEC SIRT6 deficiency impairs intestinal type 2 mucosal immunity. STAT6 is activated by type 2 cytokines and essential for anti-helminth epithelial remodeling[15]. *Stat6*[−/−] mice failed to generate helminth infection-induced tuft and goblet cell hyperplasia, demonstrating a requirement for STAT6 activity in the protective type 2 immune responses[12,17–19]. Importantly, the sufficiency of intestinal epithelial STAT6 in driving tuft and goblet cell hyperplasia has been demonstrated by IEC specific expression of constitutively activated STAT6 (STAT6vt) in two mouse models generated by Schubart *et al.* and us[20]. Therefore, STAT6 activity must be fine-tuned for proper type 2 immune responses to helminth infections. Upon IL4/IL13 stimulation, STAT6 is tyrosine phosphorylated by JAK kinases, dimerized via the SH2 domain, and then translocated to the nucleus for DNA binding and gene transcription[16]. Our data demonstrate that SIRT6 is required for STAT6 activation in IECs because rIL4/rIL13 treatment fails to rescue the impaired anti-helminth responses in intestinal epithelium of *Sirt6* IEC-

KO mice. Importantly, epithelial overexpression of STAT6vt, a constitutively active form of STAT6, markedly reverses almost all the abnormalities, such as diminished tuft and goblet cell hyperplasia and increased worm burden, observed in *H. poly*-infected IEC-KO mice, confirming the necessity of SIRT6 for proper IL4/IL13-induced STAT6 activation. Therefore, by modulating STAT6 activity, epithelial SIRT6 exert an effect on type 2 immunity-induced epithelial remodeling.

But it should be pointed out that *Sirt6* ablation in IECs also results in reduced intestinal tuft cell abundance in naive mice or in vehicle-treated organoids, suggesting epithelial SIRT6 is also required for intestinal tuft cell development at the steady-state when type 2 immunity is not activated. So far, the mechanisms that give rise to the tuft cell lineage are less well understood, and STAT6 and POU2F3 are known transcription factors to be essential for tuft cell differentiation[12,13,19]. The necessity of STAT6 for helminth-associated epithelial remodeling is supported by the findings from *Stat6*[−/−] mice[12,17–19]. However, we found comparable tuft cell frequency between intestinal organoids isolated from *Stat6*[−/−] mice and WT controls in the absence of IL4/IL13 stimulation, indicating STAT6 is not required for tuft cell differentiation at the steady-state. So, the decreased intestinal tuft cell frequency we observed in naive IEC-KO mice or vehicle-treated IEC-KO organoids is unlikely caused by the changes in STAT6 activity. POU2F3 is a master regulator and an absolute requisite for tuft cell differentiation from epithelial crypt progenitor cells[44]. *Pou2f3*[−/−] mice completely lack intestinal tuft cells and have defective mucosal type 2 immune responses to helminth infection[13]. Intriguingly, our data show that *Pou2f3* expression is also significantly reduced in the intestinal epithelium or organoids from IEC-KO mice at the steady-state. Notably, we cannot tell whether the decrease of *Pou2f3* expression is a cause or a consequence of diminished tuft cell population caused by epithelial *Sirt6* deletion. But this raises the possibility that SIRT6 may also directly modulate *Pou2f3* expression and/or activity to control tuft cell differentiation when type 2 immunity is not activated. It still needs further investigation about how epithelial SIRT6 specifically controls tuft differentiation to maintain the steady-state intestinal epithelium homeostasis.

Basically, as a histone deacetylase, SIRT6 epigenetically suppresses gene transcription[45]. To identify downstream mediators through which SIRT6 promotes the phosphorylation of STAT6 (Y641), we screened for factors known to inhibit IL4/IL13-STAT6 signaling. SOCS3 belongs to the suppressors of cytokine signaling (SOCS) family of proteins that mediate negative-feedback inhibition of the JAK−STAT pathway[46,47]. Our initial study revealed that SOCS3 expression was significantly upregulated in the IECs of naive or *H. poly*-infected *Sirt6* IEC-KO mice. Conversely, *H. poly*-infected SIRT6 IEC-Tg mice exhibited

suppressed expression of SOCS3 in IECs. Furthermore, in vitro studies performed in NCM460 cells with overexpression or knockdown of SIRT6 verified a direct control of epithelial expression of SOCS3 by SIRT6. In the gastrointestinal tract, SOCS3 is expressed by epithelial and lamina propria cells, though in homeostasis epithelial expression is normally low[48,49]. Importantly, SOCS3 is a well-known feedback inhibitor of cytokine-mediated STAT3 signaling, avoiding STAT3 phosphorylation[41]. Intriguingly, SOCS3 has also been found to play inhibitory roles towards IL4/IL13-STAT6 signaling in either immune cells or non-immune cells. In macrophages, increased SOCS3 expression is inversely correlated with suppressed p-STAT6 (Y641) levels, indicating that SOCS3 may participate in the repression of STAT6 phosphorylation[50]. Besides, Zhang, *et al.* have demonstrated that SOCS3 interacts with TYK2 and mediates its degradation, thereby leading to suppression of IL13-mediated STAT6 activation in air epithelial cells[51]. In accordance with the notion that SOCS3 inhibits STAT6 activation, we observed that SOCS3 expression was inversely related to the p-STAT6 (Y641) levels in the intestinal epithelium of mice infected with *H. poly* by different time intervals. Furthermore, ectopic expression of SOCS3 in NCM460 cells significantly reduces the phosphorylation of STAT6 (Y641), providing direct evidence for SOCS3-mediated suppression of STAT6 phosphorylation in IECs. It is worth noting that SIRT6 binds directly to transcription factors and modifies the local chromatin status to regulate gene transcription[22,23]. However, we still don't know the transcription factors through which SIRT6 regulates SOCS3 expression in IECs. Notably, it is widely accepted that SOCS3 expression is in itself regulated by STATs, especially STAT3[48]. Whether STATs participate in regulating the expression of SOCS3 together with SIRT6 needs to be clarified in future studies. Collectively, we have provided evidence that SOCS3 is a downstream effector of SIRT6 to suppress the STAT6 (Y641) phosphorylation in the intestinal epithelium.

In summary, the current study for the first time demonstrates that intestinal epithelial SIRT6 promotes the differentiation of tuft and goblet cell induced by helminth infection. Mechanistically, SIRT6 upregulates STAT6 phosphorylation and activity through epigenetically suppressing SOCS3 expression in IECs (Fig. 8v). Our findings demonstrate a requirement for IEC SIRT6 in regulating helminth infection-induced intestinal epithelial remodeling to orchestrate an effective anti-helminth immunity.

## Methods
### Animal studies
*Sirt6* floxed mice (*Sirt6^{flox/flox}*), SIRT6 transgenic mice carrying a floxed stop cassette at the Rosa 26 locus (*TgSIRT6^{flox/flox}*) were kindly provided by Dr. X.Charlie Dong at Indiana University School of Medicine[37,38]. *Villin-Cre* mice were kindly provided by Dr. Wei-Qi He at Soochow University[52]. TgSTAT6vt mice expressing the *Vil1* promoter driven-STAT6vt was generated by GemPharmatech through random insertion of the transgene into the genome. *Sirt6^{flox/flox}* mice and *TgSIRT6^{flox/flox}* mice were bred to *Villin-Cre* mice to generate IEC-KO mice and IEC-Tg mice, respectively. IEC-KO were bred to TgSTAT6vt mice to generate IEC-KO-TgSTAT6vt mice. All mice used in this study were on a C57BL/6 J genetic background. Both male and female mice at 2–4 months of age were used for experiments. Mice were housed in a temperature (22 ± 2 °C) and humidity (40–60%)-controlled facility with regular 12:12 light/dark cycle and fed a rodent chow with free access to water. All animal procedures were performed in accordance with "Guide for the Care and Use of Laboratory Animals" published by the National Institutes of Health and were approved by the Institutional Animal Care and Use Committee of Xinxiang Medical University.

### Intestinal epithelium (IECs) isolation
Freshly harvested jejunum (~3 cm) was flushed with ice-old PBS, opened longitudinally, and cut into ~0.5 cm pieces. The tissues were rotated at

4 °C in buffer A (3 mM EDTA, 2 mM DTT in PBS) for 15 min, and followed by buffer B (3 mM EDTA in PBS) for 45 min. The tissues were then vigorously shaken to release the epithelium, and the supernatants were collected and pelleted for protein or RNA extraction.

### Helminth infection&rIL4/rIL13 treatment
For *H. poly* infection, mice were infected by oral gavage with 200 *H. poly* L3 larvae and sacrificed on day 14 post infection to collect tissues for experiments or to determine worm burden. For determination of egg burden, fecal pellets were loosened in PBS and counted numbers of eggs using a McMaster counting chamber. The numbers of eggs were normalized to the weight of feces. For analysis of adult worm burden, the duodenum was cut open longitudinally and followed by incubation in 37 °C PBS for 2 h to harvest and count worm numbers using a dissection microscope. For rIL4/rIL13 treatment, starting on day 6 and ending on day 14 post-*H. poly* infection, mice were injected intraperitoneally (*i.p.*) daily with a mixture of interleukins (0.5 μg rIL4 + 0.5 μg rIL13 per mouse).

### Intestinal organoid culture
Isolated jejunal crypts were counted and cultured in Matrigel following a published protocol[53]. Growth medium consists of Advanced DMEM/F12 with GlutaMAX, HEPES, Pen/Strep, 1× N2 supplement, 1× B27 supplement, EGF (50 ng/ml), Noggin (100 ng/ml), and R-Spondin 1-conditioned medium. Organoids were maintained at 37 °C and medium was changed every 2 days. Cultures were split weekly by mechanical disruption of organoids. Organoids at passages 2–4 were used for experiments. Murine rIL13 (25 ng/ml) was added to organoid culture medium for 48 hrs to stimulate tuft and goblet cell expansion.

### Real-time RT-PCR
Total RNAs were extracted from tissues or cells using TRIzol reagent (Takara Bio) and converted into cDNAs using a cDNA synthesis kit (Vazyme). Real-time PCR analysis was performed using SYBR Green Master Mix (Vazyme) in ABI StepOnePlus Real-Time PCR system. Gene expression levels were calculated using the ΔΔCt method after their normalization to the expression levels of the housekeeping gene *Rplp0*. Sequence information of the primers used in this study is listed in Supplementary Table 1.

### RNA sequencing
Total RNAs of isolated jejunal IECs were extracted by using the RNeasy Plus Mini Kit (Qiagen) following the manufacturer's instructions. RNA concentration and integrity of each sample were measured on an Agilent Bioanalyzer. A cDNA library was prepared and sequenced according to the Illumina standard protocol by Vazyme Biotech. Sequencing reads were mapped to the mouse reference genome (mm10) using Hisat2 2.1.0 with default settings[54]. Differentially expressed genes were analyzed using counts from HTSeq-count (version 0.13.5)[55]. Only genes that have at least 10 reads in total were then fed into DESeq2 with masking structural RNAs and non-coding RNAs for calculating fold change[56]. Genes that are significantly changed were selected by using log2(fold change) > 1.25 for upregulated genes and log2(fold change) < −1.25 for downregulated genes with $p$ value < 0.05 as cut off. The gene ontology analysis was performed using Metascape for up- and downregulated genes[57]. To generate a ranked gene list for pre-ranked gene set enrichment analysis (GSEA), genes were ranked by their log2(fold-change) in IEC-KO versus LoxP ($n = 3$)[58]. The RNA-seq data were deposited at Gene Expression Omnibus (GEO) under accession ID GSE202470 (https://www.ncbi.nlm.nih.gov/geo/query/acc.cgi?acc=GSE202470).

### Western blot
Tissues and cells were lysed in RIPA buffer supplemented with 1 mM phenylmethylsulfonyl fluoride (PMSF) and Roche cOmplete protease

inhibitor cocktail. Protein extracts were separated on an SDS-PAGE gel, transferred to nitrocellulose membranes, and blotted with the indicated primary antibodies at 4 °C for overnight. Horseradish peroxidase (HRP)-conjugated secondary antibodies were given for 1 h. The immune complexes were detected using the ECL detection reagents (Beyotime). Band densitometries were obtained with ImageJ software. Detailed information of the primary antibodies used in this study is listed in Supplementary Table 2.

### Histology, immunohistochemistry, and immunofluorescence
Intestinal tissues were coiled into "Swiss rolls" and fixed in 4% paraformaldehyde (PFA), embedded in paraffin and sectioned at a thickness of 5 μm. Organoids were fixed in 4% PFA, embedded in optimal cutting temperature (OCT) compound and sectioned at a thickness of 10 μm. Antigen retrieval was performed in citrate buffer using a pressure cooker. Paraffin sections of intestinal tissues were stained with H&E staining according to standard procedures. Immunohistochemistry (IHC) analysis was performed using IHC detection kit (ZSGB Bio) following manufacture's guidelines. Alcian blue staining was carried out using Alcian blue stain kit (Vector Labs) following manufacturer's instructions. For immunofluorescence, tissue slides were blocked with 3% BSA, 0.2% Tween-20 in PBS, incubated with primary antibodies (1:100 to 1:200 dilution) overnight, and secondary antibodies (1:400 dilution) for 1 h. TUNEL assay was carried out using DeadEnd Fluorometric TUNEL System (Promega) following manufacturer's instructions. Images were taken with a Nikon Eclipse Ni-U microscope and DS-Fi2 color CCD (Nikon) through ImageView software. For cell number quantification, 3–5 random fields per sample were captured. In the intestine, cell number was normalized to the number of crypt/villus units. In the organoid, cell number was normalized to nuclei number as indicated by DAPI staining. Detailed information of the primary antibodies used in this study is listed in Supplementary Table 2.

### ELISA
Jejunum (~2 cm) was removed, flushed with saline, cut into several pieces and then lysed in RIPA buffer using Dounce homogenizer. IL13 levels in jejunal tissue lysate were analyzed using Mouse IL-13 DuoSet ELISA (R&D Systems) according to manufacture's protocol.

### Flow cytometry
Intestinal LP lymphocytes were isolated according to an established protocol[59]. Briefly, freshly harvested jejunum (~10 cm) was flushed, opened longitudinally, and cut into ~5–6 pieces. DTT and EDTA solutions were used sequentially to remove intraepithelial lymphocytes and epithelial cells. The remaining tissue was washed and digested with collagenase for 45 min at 37 °C with shaking. LP lymphocytes were pelleted after filtering the digested tissue through 100 μm filters. Cells were fixed and then stained with relevant antibodies at 4 °C for 30 min. Flow cytometry data were acquired on CytoFLEX (Beckman Coulter) and analyzed with FlowJo (v10.6.2). When gating cells, single cells were selected for viable, lineage negative (CD19⁻CD11b⁻CD49b⁻CD11c⁻Gr-1⁻B220⁻), CD45⁺ cells. Within the CD3⁻CD4⁻ subset, ILC2 population is positive for CD90 and IL-17RB. Detailed information of the antibodies used in this study is listed in Supplementary Table 2.

### Chromatin immunoprecipitation (ChIP)
NCM460 cells were fixed with 1% formaldehyde at RT for 15 min. Chromatins were lysed and sonicated to an average size of ~250 bp. Immunoprecipitation was performed using anti-FLAG and anti-Acetyl-H3K56 antibodies at 4 °C overnight. Reversion of cross-linking was performed by heating samples at 65 °C for overnight and DNA was purified using phenol-chloroform extraction. ChIP DNA amount for gene promoters of interest was analyzed by real-time PCR and normalized to that of a housekeeping gene, *ACTB*. Sequence information of the primers used in this study is listed in Supplementary Table 1.

### Luciferase reporter assay
HEK293T cells were plated onto 24-well plates and co-transfected with p2xSTAT6-Luc2P (luciferase reporter), PGL4.74 (Renilla luciferase), and expression constructs. Transfection was conducted using Lipofectamine 2000 (Invitrogen). 48 h after transfection, cells were treated with vehicle or rIL13 for 2 h. Then, cells were lysed and luciferase activity was measured using the Dual-Luciferase Reporter Assay System and GloMax™ Systems (Promega). Data were presented as relative Firefly luciferase activities normalized to Renilla luciferase activities.

### Statistics and reproducibility
All data are presented as mean ± SEM. Data were analyzed using two-tailed unpaired Student's *t*-test. $p < 0.05$ was considered as significant. If not otherwise mentioned, each experiment was repeated independently with similar results at least three times.

### Reporting summary
Further information on research design is available in the Nature Research Reporting Summary linked to this article.

## Data availability
RNA-sequencing data generated during this study have been deposited in the Gene Expression Omnibus (GEO) under accession number GSE202470 (https://www.ncbi.nlm.nih.gov/geo/query/acc.cgi?acc=GSE202470). The authors declare that all other data supporting the findings of this study are available from the corresponding author upon reasonable request. Source data are provided with this paper.

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

## Acknowledgements

We thank Ms. Xiyu Yang for animal husbandry and Dr. Yinming Liang for helpful discussions on flow cytometry. This work was supported by

grants from Program for Science & Technology Innovation Talents in Higher Education of Henan Province (20HASTIT046 to X.X.), National Natural Science Foundation of China (U1904132 to X.X.), Key Scientific and Technological Project of Xinxiang (GG2019008 to Q.W.), Outstanding Youth Foundation of Jiangsu Province (BK20190043 to W.-Q.H.), International Joint Research Center for Genomic Resources (2017B01012 to W.-Q.H.), National Institute of Health/National Institute of Allergy and Infectious Diseases (R01 AI162791 to H.-B.R.), Key Scientific and Technological Research Project in Henan Province (222102310048 to G.Z.).

## Author contributions

X.X. conceived the hypothesis, designed and performed the experiments, analyzed data, and wrote the manuscript; C.Y., W.-Q.H., J.Y., Y.X., and X.Z. performed experiments and analyzed data; R.H., H.M., S.X., Z.L., J.M., L.X., and Q.W. assisted with the experiments and data collection; K.R. provided the larvae (L3) of *H. poly* helminth; X.S.W. and C.R.V. analyzed the RNA-seq data; J.Z. provided NCM460 cells; G.Z. and X.Z. assisted with flow cytometry; Y.S. assisted with ChIP assay; H.-B.R. generated TgSTAT6vt transgenic mice; Q.W. conceived the hypothesis, designed the experiments, analyzed and interpreted the data, and revised the manuscript.

## Competing interests

C.R.V. has received consulting fees from Flare Therapeutics, Roivant Sciences, and C4 Therapeutics; has served on the scientific advisory board of KSQ Therapeutics, Syros Pharmaceuticals, and Treeline Biosciences; has received research funding from Boehringer-Ingelheim and Treeline Biosciences; and has a stock option from Treeline Biosciences. All other authors declare no competing interests.

## Additional information

[1]School of Forensic Medicine, Xinxiang Medical University, Xinxiang, Henan 453003, China. [2]Xinxiang Key Laboratory of Metabolism and Integrative Physiology, Xinxiang Medical University, Xinxiang, Henan 453003, China. [3]Jiangsu Key Laboratory of Neuropsychiatric Diseases and Cambridge-Suda (CAM-SU) Genomic Resource Center, Suzhou Medical College of Soochow University, Suzhou, Jiangsu 215123, China. [4]School of Basic Medical Sciences, Xinxiang Medical University, Xinxiang, Henan 453003, China. [5]School of Laboratory Medicine, Xinxiang Medical University, Xinxiang, Henan 453003, China. [6]The Third Affiliated Hospital of Xinxiang Medical University, Xinxiang, Henan 453003, China. [7]Hunan Normal University School of Medicine, Changsha, Hunan 410013, China. [8]Cold Spring Harbor Laboratory, Cold Spring Harbor, NY 11724, USA. [9]School of Pharmacy, Xinxiang Medical University, Xinxiang, Henan 453003, China. [10]Department of Integrative Biology and Physiology, University of Minnesota Medical School, Minneapolis, MN 55455, USA. ✉e-mail: xwxiong@xxmu.edu.cn; wqzmgl@126.com

