## [Peer Review File · Nature Communications]

Reviewer comments, first round review

REVIEWER COMMENTS

Reviewer #1 (Remarks to the Author):

In this manuscript, Xiong and collaborators address the function of Sirt6 gene in the type 2 immune response-triggered epithelial remodelling of the small intestine. Using both loss- and gain-of-function mouse models, as well as intestinal organoid culture, they demonstrate that Sirt6 is required for an anthelmintic response, in an epithelium autonomous way. Their findings show that Sirt6 protein is an upstream regulator of the Stat6 transcription factor in response to IL13, through suppression of Socs3 expression. The manuscript is well-written, and experiments are generally well-designed, including adequate controls. My overall feeling is that this study might potentially be interesting for a broad readership after being improved by additional experiments and explanations.

The expression patterns of both Sirt6 and Stat6 (both total and phosphorylated forms) should be shown more accurately. From the data shown by the authors, it is difficult to appreciate the actual nature of the cells that express these proteins, as well as their sub-cellular localisation. Authors should definitively improve this, showing: (1) higher magnification images, and (2) describing both expression patterns during the full course of infection time points (naïve mice, 6 dpi, 9 dpi, 14 dpi, and a later time point after worms expulsion).

Although data generated from organoid experiments strongly point towards an epithelial autonomous effect of Sirt6 deficiency, decrease of the type 2-related interleukins shown in figure 2 may indicate that immune cells also take a part in the H.Poly susceptibility phenotype. Additional experiments should be added to clarify this point. First, authors should confirm their finding (e.g. decrease of type-2 interleukins), using alternative methods based on proteins detection (ELISA or CBA). Then, in vivo rescue experiments, treating infected Sirt6-deficient mice with IL13, would definitively help to figure out to what extent immune cells activation defects contribute to this phenotype. Consequently, the discussion part of the manuscript may also require edition.

In some of the panels of the figure 2, the authors must show data from naïve mice (figure 2C, 2E, 2H and 2I).

The scheme proposed by the authors in figure 8 is quite useful, but seems to be wrong regarding the architecture of the tissue: from my interpretation, worms are likely to be drawn within the lamina propria and immune cells in the lumen. Could the authors confirm, or not, my feeling, and, if necessary, edit this scheme?

Reviewer #2 (Remarks to the Author):

This is an interesting study to interrogate the role of Sirt6 in worm-induced type 2 immunity. Using loss-of-function and gain-of function mouse models, intestinal organoids, and heterologous expression, the authors built a strong case that Sirt6 modulates worm-induced type 2 immunity by activating epithelial Stat6 activity, which is a master transcription factor for the development of type 2 immune responses. Mechanistically, they found that Sirt6 decreases Socs3 expression, which negatively regulate the activity of STAT6 by attenuating phosphorylation of Stat6. Overall, this study provides new insight into type 2 immunity, but there are some concerns that need to be addressed.

1. The title of the manuscript is "SIRT6 promotes helminth-induced tuft and goblet cell hyperplasia through activating epithelial STAT6 activity". However, it is clear that tuft and goblet cell homeostasis in naïve mice or organoids treated with vehicle is also affected by epithelial deletion

(Figs. 1 & 3A) of Sirt6. Therefore, Sirt6 may not only affect worm-induced type 2 immunity but also affect tuft and goblet cell homeostasis. This should be discussed in the manuscript.

2. In the connecting of dots between Sirt6, Socs3 and Stat6, the authors showed that overexpression of Socs3 can attenuates phosphorylation of STAT6 in 293T and NCM460 cells (Fig. 8M, N) and that overexpression of Sirt6 can significantly enhanced the transcriptional activity of STAT6 in the absence of presence of IL13 stimulation (Fig. 4G). However, whether Socs3 is required for mediating the effect of Sirt6 on Stat6 has not been directly addressed. Would knockout or knockdown of Socs3 blunt the effect of Sirt6, for instance in HEK293T cells overexpressing Stat6, Sirt6, and p2XStat6-Luc reporter? Would knockdown or knockout of Socs3 leads to increase in P-Stat6 in NCM460 cells? These experiments are relatively easy to perform to establish the necessity of Socs3 in the proposed model. Obviously, a knockout mouse model of Socs3 would provide more convincing evidence to establish the necessity of Socs3 in the regulation of Stat6 by Sirt6. But HEK293T data would suffice to address this concern. Establishing the necessity of Socs3 is critical for the proposed mechanism.

3. The authors showed that Sirt6 modulates the activity of STAT6 via Socs3, however, it is not clear where this process take places, presumably it take place in intestinal stem cells. The authors may want to clarify this. Based on immunostaining (Fig. S4), it appears crypt cells express Sirt6. Nevertheless, cells of the villi also showed Sirt6 immunoreactivity (low resolution makes it hard to distinguish). What types of cells are these Sirt6 immuno-positive cells of the villi? Because tuft cells, ILC2, intestinal stem cells constitute a feedback circuit for mediating worm-induced type 2 immunity, any disturbance of this circuit can lead to attenuated or abolished type 2 responses. Should Sirt6 is expressed in tuft cells, there is a possibility that the initiation of type 2 immunity by tuft cells is affected in IEC-KO. The authors may want to discuss this possibility as it may also explain attenuated type 2 responses observed in IEC-KO, consistent with the reduced expression of type 2 cytokines (Fig. 2I). As a matter of fact, increase in the expression of tuft cell markers or goblet cell markers was quite dramatic in organoids generate from IEC-KO mice in response to IL-13 compared to vehicle treatment (Fig. 3A), suggesting the response to IL-13 is largely conserved in IEC-KO mice. The slightly lower expression of tuft cell markers in IL-13 treated IEC-KO organoids compared to Loxp organoids could be due to the low expression of tuft cell markers at homeostasis (e.g., altered cell fate determination that may or may not be unrelated to Stat6) (Fig. 3A). Again, Sirt6 regulates tuft and goblet cell homeostasis in both homeostasis and parasitic infection. Please see the point #1.

4. Fig3A, it is unclear if biological or technical replicates were used for qPCR analysis. If technical replicates, biological replicates are needed. Please indicate the number of samples used as well.

5. In organoids treated with IL-13, dose IEC ablation of Sirt6 reduce Tyr641 phosphorylation of Stat6 (related to Fig. 4)?

6. Fig. 3D & E, please double check labels of x-axis, IEC-KO-Veh and Loxp-IL-13 appear to be switched.

7. TgSTAT6vt mice have a very interesting phenotype (e.g., dramatic reduction of jejuna villus). A recent report (Xi et al., PNAS, 2021 118 (30) e2026307118) showed that type upregulation of Gsdmcs is associated with accelerated lytic cell death in gut epithelium. Could this explain the dramatic reduction of villus height? It would be interesting to see if Gsdmcs are upregulated in the Stat6 transgenic mice.

8. Figure 7. Lack of data on naïve IEC-Tg mice.

Reviewer #3 (Remarks to the Author):

Overview

This is an interesting paper reporting a new role for Sirt6 in the type 2/STAT6-dependent epithelial response to helminth infection and succinate stimulation. While the manuscript is technically good (although a number of questions are raised below), its novelty and precision are less convincing. The requirement for STAT6 in epithelial cells has been known for a long time, and the authors shed little new light on STAT6 itself as they only measure Y641 phosphorylation (see #1 below). The involvement of Sirt6 (which is not a protein kinase) is indeed novel and much of the data on this is clear, but its role is not at all absolute and there is no broad scope to address its downstream targets eg by RNAseq or proteomics; instead the authors select SOCS3 as a likely intermediary. Here the data do point in the right direction but the hypothesis is not fully validated and other possibilities not explored.

Major Comments

1. Regarding STAT6, only Y641 phosphorylation is measured; however there may also be serine phosphorylation and methylation; other STAT proteins such as STAT3 can also be acetylated. Given that Sirt6 is not a protein kinase but does have deacetylase and other catalytic properties, it would seem logical to assess other modifications to STAT6 eg by proteomics in the WT and KO settings.
2. In Figure 2, while tuft and goblet cell numbers are clearly lower in infected KO mice compared to WT, there is no uninfected KO control, so we cannot tell whether there is a response in the KO but from a lower baseline. In fact, this scenario appears to be in effect in the organoids, eg for Dclk1 the increment of IL-13/Vehicle is actually higher in the KO than in the WT.
3. The rationale for selecting SOCS3 as a likely target for Sirt6 repression is reasonable, but there are likely to be other players; it would seem better to conduct an unbiased RNAseq or proteomic comparison of the WT and KO IECs to identify a fuller range of target molecules. As the paper stands, they do not show data to justify the statement in the Abstract "Mechanistically, SIRT6 ablation induces SOCS3 expression" (which in any case is not a logical statement as SOCS3 expression occurring in the absence of SIRT6 must be induced by something else).
4. Key experimental data are missing from the Figure legends; first, while the Methods section states both male and female mice were used, none of the legends specify which sex were used. This is important because of the report that Sirt6 has a sex-specific effect (on lifespan in male mice only). Secondly, while group sizes are given for the experiments presented there is no statement of the number of times each experiment was repeated with similar results.
5. Generally, there are many instances where small differences, or even differences that do not reach statistical significance, are given great weight. For example, in Figure 1 C, D, I or J, none of the data on goblet cells (in or their genes) attain statistical significance. In the same Figure there is total ablation of SIRT6 in KO but really modest effect on tuft cells – in F perhaps 25% reduction. Similarly, in Figure 7B the increment in Sirt6 in the transgenic model is massively higher than increment in P-STAT6, and there are only marginal cellular changes in D, J and K. These results imply (a) that SIRT6 is a secondary modifier rather than an inducer; and (b) that its effects may be dependent on the concentrations of the primary inducers. This could be tested by titrating IL-13 in the organoid system.
6. What is the rationale for sampling the jejunum in all experiments, even those with H polygyrus – a parasite that inhabits the duodenum?

7. In Figure 8 M-O, the authors show that SOCS3 inhibits P-STAT in two cell lines, but do not use this system to analyze Sirt6; increasing induction of Sirt6 should restore P-STAT6 in a titratable manner.

Minor Comments

8. Line 67 secrete

9. Line 116 – is twofold “remarkably reduced”?

10. Line 181 states 150 mM succinate; figure legend (and cited reference) states 100 mM

11. Figure 4 A. At 9 days post infection, 2 mice are strongly positive, 1 is negative. However the quantification shown in Figure 4B has a very small standard error. How can this be?

12. Figure 4 C shows significant P-STAT6 in the steady state, not consistent with blot in A that has no staining.

13. Figure 4 G The 293T reporter assay is not explained in legend

14. The Discussion is essentially 3 very long paragraphs, over a page each, which makes it difficult for the reader to follow.

We would like to thank the reviewers for their constructive comments on our manuscript. Those comments are very helpful for revising and improving our paper. We have studied the comments very carefully and made all efforts to address them. All revised portions are highlighted in yellow color. The following is our point-by-point responses.

Reviewer #1:

In this manuscript, Xiong and collaborators address the function of Sirt6 gene in the type 2 immune response-triggered epithelial remodeling of the small intestine. Using both loss and gain-of-function mouse models, as well as intestinal organoid culture, they demonstrate that Sirt6 is required for an anthelmintic response, in an epithelium autonomous way. Their findings show that Sirt6 protein is an upstream regulator of the Stat6 transcription factor in response to IL13, through suppression of Socs3 expression. The manuscript is well-written, and experiments are generally well-designed, including adequate controls. My overall feeling is that this study might potentially be interesting for a broad readership after being improved by additional experiments and explanations.

1.The expression patterns of both Sirt6 and Stat6 (both total and phosphorylated forms) should be shown more accurately. From the data shown by the authors, it is difficult to appreciate the actual nature of the cells that express these proteins, as well as their subcellular localization. Authors should definitively improve this, showing: (1) higher magnification images, and (2) describing both expression patterns during the full course of infection time points (naïve mice, 6 dpi, 9 dpi, 14 dpi, and a later time point after worms expulsion).

Response: Thanks for your suggestions. We have conducted SIRT6 and P-STAT6 (Y641) immunostaining for jejunum tissues of naïve mice and mice infected with *H.poly* by different time points (6 dpi, 9 dpi, and 14 dpi), and included the images

(magnification at 200x and 400x) in the revised manuscript (Figure 4a&b). Consistent with our WB data, immunostaining also revealed that *H.poly* infection elevates SIRT6 expression and Y641 phosphorylation of STAT6, which are mainly localized in the nucleus of IECs. SIRT6 expression does not limit to specific cell types in intestinal epithelium. This is also confirmed by the published scRNA-seq data (PMID: 29144463) (Figure shown below). As an extensively expressed HDAC, SIRT6 plays different roles in different cell types through interacting with specific co-factors. So, it is reasonable to speculate that SIRT6 may interact with other unknown factors to exert its function on specific regulation of tuft cell differentiation in intestinal epithelium.

For STAT6 (total) immunostaining, we tested 2 antibodies from different vendors (Abcam, ab32520 & Proteintech, 51073-1-AP). Unfortunately, we could not get reliable positive signals from STAT6 immunostaining. But from WB and qPCR that examine the levels of STAT6 protein and *Stat6* mRNA (Figure shown below), we noticed that *H.poly* infection did not alter the expression of STAT6 in jejunum. It has to be pointed

out that it takes about 2 months for mice to completely clear the *H.poly* infection, so we did not perform immunostaining at a later time point when worms are expelled. But, we speculate that STAT6 phosphorylation will return to the basal level when the *H.poly* infection is cleared.

2. Although data generated from organoid experiments strongly point towards an epithelial autonomous effect of *Sirt6* deficiency, decrease of the type 2-related interleukins shown in figure 2 may indicate that immune cells also take a part in the *H.Poly* susceptibility phenotype. Additional experiments should be added to clarify this point. First, authors should confirm their finding (e.g. decrease of type-2 interleukins), using alternative methods based on proteins detection (ELISA or CBA). Then, *in vivo* rescue experiments, treating infected *Sirt6*-deficient mice with IL13, would definitively help to figure out to what extent immune cells activation defects contribute to this phenotype. Consequently, the discussion part of the manuscript may also require edition.

Response: We have done IL13 ELISA to examine the IL13 protein levels in the jejunum of IEC-KO/LoxP mice and IEC-Tg/WT mice, as shown in Fig. 1n and Fig. 7g, respectively. We found that, after *H.poly* infection, jejunal IL13 protein levels were reduced in IEC-KO mice while elevated in IEC-Tg mice compared with their corresponding control mice.

To figure out to what extent type 2 immune cells activation defects contribute to the phenotype we observed in *H.poly* infected IEC-KO mice, we followed reviewer's suggestion and treated infected mice with recombinant IL4&13 for 8 days (started on 6 dpi after *H.poly* L3 larvae inoculation and ended on 14 dpi, 1ug interleukins/mouse/day). rIL4&13 treatment rescued the impaired ability to expel *H.poly* in IEC-KO mice. Intriguingly, we still observed diminished tuft and goblet cell

hyperplasia after rIL4&13 treatment, suggesting that *Sirt6* ablation results in IEC-autonomous defects in response to type 2 cytokines (Fig. 2). These data provide further evidence that SIRT6 in IECs modulates the activation of IL4/13-STAT6 pathway to control the epithelial remodeling in response to helminth infection.

Moreover, we re-edited the discussion section following the reviewers' suggestions. For example, we discussed that SIRT6 may regulate tuft cell differentiation through IL4/13-STAT6 independent pathways in the naïve state. We also discussed the results we obtained from experiments of rIL4&13 treatment.

3. In some of the panels of the figure 2, the authors must show data from naïve mice (figure 2C, 2E, 2H and 2I).

Response: Thank you for your reminding. We have combined the data from old Figures 1 (naïve) & 2 (*H.poly* infected) and made a revised Figure 1. It is worthy to note that we only show the qPCR data of jejunal type 2 cytokine expression from *H.poly*-infected mice because we could not get reliable qPCR results from naïve mice due to their low expression levels (Fig. 1m).

4. The scheme proposed by the authors in figure 8 is quite useful, but seems to be wrong regarding the architecture of the tissue: from my interpretation, worms are likely to be drawn within the lamina propria and immune cells in the lumen. Could the authors confirm, or not, my feeling, and, if necessary, edit this scheme?

Response: We thank you for pointing this out. Indeed, the architecture of the intestine in the scheme in Fig. 8 was not correct. As shown in revised Fig. 8v, we have re-edited this scheme accordingly.

Reviewer #2:

*This is an interesting study to interrogate the role of *Sirt6* in worm-induced type 2 immunity. Using loss-of-function and gain-of function mouse models, intestinal*

organoids, and heterologous expression, the authors built a strong case that Sirt6 modulates worm-induced type 2 immunity by activating epithelial Stat6 activity, which is a master transcription factor for the development of type 2 immune responses. Mechanistically, they found that Sirt6 decreases Socs3 expression, which negatively regulate the activity of STAT6 by attenuating phosphorylation of Stat6. Overall, this study provides new insight into type 2 immunity, but there are some concerns that need to be addressed.

1. The title of the manuscript is “SIRT6 promotes helminth-induced tuft and goblet cell hyperplasia through activating epithelial STAT6 activity”. However, it is clear that tuft and goblet cell homeostasis in naïve mice or organoids treated with vehicle is also affected by epithelial deletion (Figs. 1 & 3A) of Sirt6. Therefore, Sirt6 may not only affect worm-induced type 2 immunity but also affect tuft and goblet cell homeostasis. This should be discussed in the manuscript.

Response: Thanks for your nice comments. Actually, epithelial *Sirt6* deletion has no effect on the frequency of goblet cells in naïve mice or vehicle-treated organoids (Fig. 1&Fig. 3). But, as we showed in our manuscript (Fig.1&Fig.3), SIRT6 does regulate tuft cell homeostasis at the steady state. In naïve mice, due to tonic type 2 immune signaling, pY-STAT6 activity can still be detected by western blot (Fig. 4f) and immunostaining (Fig. 4b, e). It is possible that SIRT6, maybe partly, regulates tuft cell development through STAT6 at the naïve state. Based on the findings that we also found reduced tuft cell abundance in vehicle-treated KO organoids (almost no STAT6 activity due to the lack of type 2 cytokines in culture system), we agree with the reviewer that STAT6-independent mechanisms may exist (Please refer to the responses to Question 3). Accordingly, we changed the manuscript title to “Sirtuin 6 maintains epithelial STAT6 activity to support intestinal tuft cell development and type 2 immunity” and revised the discussion section.

2. In the connecting of dots between Sirt6, Socs3 and Stat6, the authors showed that overexpression of Socs3 can attenuates phosphorylation of STAT6 in 293T and

NCM460 cells (Fig. 8M, N) and that overexpression of Sirt6 can significantly enhanced the transcriptional activity of STAT6 in the absence or presence of IL13 stimulation (Fig. 4G). However, whether Socs3 is required for mediating the effect of Sirt6 on Stat6 has not been directly addressed. Would knockout or knockdown of Socs3 blunt the effect of Sirt6, for instance in HEK293T cells overexpressing Stat6, Sirt6, and p2XStat6-Luc reporter? Would knockdown or knockout of Socs3 leads to increase in P-Stat6 in NCM460 cells? These experiments are relatively easy to perform to establish the necessity of Socs3 in the proposed model. Obviously, a knockout mouse model of Socs3 would provide more convincing evidence to establish the necessity of Socs3 in the regulation of Stat6 by Sirt6. But HEK293T data would suffice to address this concern. Establishing the necessity of Socs3 is critical for the proposed mechanism.

Response: In the revised manuscript, we followed the reviewer's suggestions and conducted additional western blot and luciferase assay experiments to confirm the hypothesis that SIRT6 is dependent on SOCS3 to regulate STAT6 Y641 phosphorylation (Fig.8p-u). Our new data show that SOCS3 is required for SIRT6 to regulate STAT6 phosphorylation and activity in NCM460 cells or HEK293T cells (with ectopic expression of STAT6).

3. The authors showed that Sirt6 modulates the activity of STAT6 via Socs3, however, it is not clear where this process take places, presumably it take place in intestinal stem cells. The authors may want to clarify this. Based on immunostaining (Fig. S4), it appears crypt cells express Sirt6. Nevertheless, cells of the villi also showed Sirt6 immunoreactivity (low resolution makes it hard to distinguish). What types of cells are these Sirt6 immuno-positive cells of the villi? Because tuft cells, ILC2, intestinal stem cells constitute a feedback circuit for mediating worm-induced type 2 immunity, any disturbance of this circuit can lead to attenuated or abolished type 2 responses. Should Sirt6 is expressed in tuft cells, there is a possibility that the initiation of type 2 immunity by tuft cells is affected in IEC-KO. The authors may want to discuss this possibility as it may also explain attenuated type 2 responses observed in IEC-KO, consistent with the reduced expression of type 2 cytokines (Fig. 2I). As a matter of fact, increase in the

expression of tuft cell markers or goblet cell markers was quite dramatic in organoids generate from IEC-KO mice in response to IL13 compared to vehicle treatment (Fig. 3A), suggesting the response to IL13 is largely conserved in IEC-KO mice. The slightly lower expression of tuft cell markers in IL13 treated IEC-KO organoids compared to Loxp organoids could be due to the low expression of tuft cell markers at homeostasis (e.g., altered cell fate determination that may or may not be unrelated to Stat6) (Fig. 3A). Again, Sirt6 regulates tuft and goblet cell homeostasis in both homeostasis and parasitic infection. Please see the point #1.

Response: Our immunostaining results demonstrate that SIRT6 is broadly expressed in intestinal epithelial cells. We also analyzed the published single cell RNA-seq data and found that *Sirt6* is universally expressed in almost all types of IECs (Please refer to Reviewer#1's 1st Question). Our data provided evidence that in IECs, SIRT6, by enhancing the phosphorylation and activation of STAT6, promoting type 2 immunity induced intestinal epithelial remodeling. In this process, we speculate the major cell type SIRT6 influences is intestinal progenitor cells (ISCs or TA cells). When type 2 immunity is activated by helminth infection, SIRT6 upregulates the Y641 phosphorylation of STAT6, then activated STAT6 promotes the progenitor cells to differentiate into tuft and goblet cells. It is still unclear if SIRT6 in mature tuft cells are sufficient to drive the anti-helminthic circuit. Future experiments are warranted in the lab to generate tuft cell-specific SIRT6 knockout and/or overexpression mice to answer this question.

To determine the dependency on STAT6, we examined the *Dclk1* (tuft cell marker) and *Retnlb* (goblet cell marker) mRNA expression in intestinal organoids from *Stat6*^{-/-} mice and WT control mice. IL4-induced tuft and goblet cell hyperplasia was totally absent in *Stat6*^{-/-} organoids (almost no increment of *Dclk1* and *Retnlb* expression), indicating STAT6 is absolutely required for type 2 immunity-induced epithelium remodeling. Intriguingly, we observed comparable *Dclk1* and *Retnlb* expression between vehicle-treated *Stat6*^{-/-} and WT organoids (Figure shown below), suggesting that STAT6 is not necessary for stochastic tuft cell marker expression in organoids. In contrast, *Sirt6* deletion led to reduced tuft cell number in naïve mice or in vehicle-

treated organoids (almost no STAT6 activity due to the lack of type 2 cytokines in culture system), which suggests that epithelial SIRT6 may regulate tuft cell differentiation through both STAT6-dependent and -independent mechanisms. Nonetheless, STAT6 phosphorylation and activity were impaired in SIRT6 IEC-KO mice, while STAT6^{vt} overexpression was sufficient to rescue the defects of tuft cell hyperplasia and anti-helminth activity in IEC-KO mice, indicating that STAT6 signaling is a major downstream target of SIRT6.

H. poly infection-induced tuft and goblet cell expansion was significantly compromised in IEC-KO mice (Fig. 1); however, as the reviewer pointed out, IL13-induced expression of tuft and goblet cell markers in IEC-KO organoids was substantial, even though still to a much lower extent compared to wildtype organoids. We postulated that a high concentration of IL13 (25 ng/ml) was used in organoids, thus masking the effect of SIRT6. Therefore, we cultured IEC-KO and LoxP organoids and treated them with different concentrations (0-5 ng/ml) of IL13 for 48 hrs. As shown in the following figure, fold increases of *Dclk1* expression in response to low concentration of IL13 was compromised in IEC-KO organoids (4.58 fold (KO) vs. 7.11 fold (WT) at 0.5 ng/ml of IL13 and 8.22 fold (KO) vs. 11.04 fold (WT) at 1 ng/ml of IL13). At higher concentrations (>2 ng/ml) of IL13, fold increases of *Dclk1* expression to vehicle treatment was comparable between genotypes, despite the fact that IEC-KO still much lower relative expression. This data indicate that high levels of IL13 may elicit STAT6- and SIRT6-independent mechanisms to promote tuft cell differentiation. It is also worth noting that SIRT6 is a HDAC, not a transcription factor, thus may function as a modifier,

not an inducer to regulate cellular processes.

4. Fig3A, it is unclear if biological or technical replicates were used for qPCR analysis. If technical replicates, biological replicates are needed. Please indicate the number of samples used as well.

Response: We have put the number of samples into figure legend of Fig. 3a.

5. In organoids treated with IL13, dose IEC ablation of Sirt6 reduce Tyr641 phosphorylation of Stat6 (related to Fig. 4)?

Response: We conducted WB to analyze the IL13 stimulated STAT6 (Y641) phosphorylation levels in IEC-KO/LoxP organoids and TgSTAT6vt/WT organoids. These WB data were added into the revised Fig. 4h&i and Fig. 7n&o.

6. Fig. 3D & E, please double check labels of x-axis, IEC-KO-Veh and Loxp-IL13 appear to be switched.

Response: Thanks for your careful review. We have made the corrections accordingly.

7. TgSTAT6vt mice have a very interesting phenotype (e.g., dramatic reduction of jejuna villus). A recent report (Xi et al., PNAS, 2021 118 (30) e2026307118) showed that type upregulation of Gsdmcs is associated with accelerated lytic cell death in gut epithelium. Could this explain the dramatic reduction of villus height? It would be interesting to see if Gsdmcs are upregulated in the Stat6 transgenic mice.

Response: Yes, *Gsdmc* genes are direct targets of STAT6. In our recent Immunity paper (PMID: 35385697), we demonstrated that *Gsdmc* expression is absent in *Stat6*^{-/-} mice while upregulated in TgSTAT6vt mice. However, we do not believe that helminth infection-induced GSDMC expression leads to pyroptosis of intestinal epithelial cells. Rather, we provided strong evidence supporting the notion that GSDMC pores formed on intestinal epithelial cells promote the unconventional secretion of IL-33. We are currently in the lab investigating how STAT6 controls intestinal stem cell function and villus/crypt structure, independent of lytic cell death.

8. *Figure 7. Lack of data on naïve IEC-Tg mice.*

Response: The data of epithelial phenotype of naïve IEC-Tg and control WT mice were put into revised Fig. S9. Notably, we did not find any differences of tuft and goblet cell abundance in intestinal epithelium of IEC-Tg and WT mice, suggesting SIRT6 is required but not sufficient for tuft cell differentiation at the steady state.

Reviewer #3:

Overview

This is an interesting paper reporting a new role for Sirt6 in the type 2/STAT6-dependent epithelial response to helminth infection and succinate stimulation. While the manuscript is technically good (although a number of questions are raised below), its novelty and precision are less convincing. The requirement for STAT6 in epithelial cells has been known for a long time, and the authors shed little new light on STAT6 itself as they only measure Y641 phosphorylation (see #1 below). The involvement of Sirt6 (which is not a protein kinase) is indeed novel and much of the data on this is clear, but its role is not at all absolute and there is no broad scope to address its downstream targets eg by RNAseq or proteomics; instead the authors select SOCS3 as a likely intermediary. Here the data do point in the right direction but the hypothesis is not fully validated and other possibilities not explored.

Major Comments

1. Regarding STAT6, only Y641 phosphorylation is measured; however there may also be serine phosphorylation and methylation; other STAT proteins such as STAT3 can also be acetylated. Given that Sirt6 is not a protein kinase but does have deacetylase and other catalytic properties, it would seem logical to assess other modifications to STAT6 eg by proteomics in the WT and KO settings.

Response: Thanks for your comments. Tyr641 of STAT6 is the key phosphorylation site required for responsiveness to IL4/IL13 stimulation. Importantly, accumulating evidence identify several sites for serine phosphorylation of STAT6 and reveal functional roles of STAT6 serine phosphorylation in IL4/IL13-mediated gene expression and other IL4/13 regulated events (PMID: 11164892, 15069079& 21123173). Moreover, methylation of STAT6 also modulates STAT6 phosphorylation, nuclear translocation, and DNA-binding activity (PMID: 15153491). Since SIRT6 mainly functions as a deacetylase, in our manuscript, we just focused our investigation on whether SIRT6 influences STAT6 acetylation or regulates gene transcriptional activity through modulating histone acetylation. As shown in Fig. S10, SIRT6 has no effect on STAT6 acetylation. In addition, we followed the reviewer's suggestion and examined whether SIRT6 influences the serine/threonine phosphorylation of STAT6 in NCM460 cells. The Ser/Thr phosphorylation of ectopically expressed STAT6-FLAG was detected using immunoprecipitation (IP) with anti-FLAG antibody followed by western blot with anti-phospho-Ser/Thr antibody (CST#9631). As shown in the following figure, after IL13 stimulation, we observed the presence of Ser/Thr phosphorylation in STAT6. However, it seems that SIRT6 overexpression did not affect STAT6 Ser/Thr phosphorylation. It should be pointed out that SIRT6 has several catalytic activities such as deacetylation, diacylation and ribosylation, allowing the regulation of a variety of signaling pathways. Thus, it is reasonable to postulate that SIRT6 may regulate epithelium homeostasis through other modifications of STAT6 or even through other mechanisms. Future studies, i.e., proteomics analysis, are warranted to elucidate these issues.

2. In Figure 2, while tuft and goblet cell numbers are clearly lower in infected KO mice compared to WT, there is no uninfected KO control, so we cannot tell whether there is a response in the KO but from a lower baseline. In fact, this scenario appears to be in effect in the organoids, eg for *Dclk1* the increment of IL13/Vehicle is actually higher in the KO than in the WT.

Response: Thanks. We have combined the data from old Figures 1 (naïve) & 2 (*H.poly* infected) and made a revised Figure 1. From the revised Figure 1, we can see that the helminth-infection induced tuft and goblet cell hyperplasia is compromised in IEC-KO mice. As the reviewer pointed out, the increment of *Dclk1* expression of IL13/vehicle is even higher in the KO than in the WT. This is the same point as Question 3 from Reviewer#2, and is addressed in my response there.

3. The rationale for selecting *SOCS3* as a likely target for *Sirt6* repression is reasonable, but there are likely to be other players; it would seem better to conduct an unbiased RNAseq or proteomic comparison of the WT and KO IECs to identify a fuller range of target molecules. As the paper stands, they do not show data to justify the statement in the Abstract “Mechanistically, *SIRT6* ablation induces *SOCS3* expression” (which in any case is not a logical statement as *SOCS3* expression occurring in the absence of *SIRT6* must be induced by something else).

Response: Thanks for your constructive suggestions. We performed RNAseq for IECs from both IEC-KO/LoxP and TgSTAT6^{vt}/WT naïve mice. The RNAseq analysis data are shown in Supplementary Fig. S7. Importantly, Gene Set Enrichment Analysis

(GSEA) revealed that tuft cell identity gene sets were enriched in downregulated genes by SIRT6 deficiency, again supporting that epithelial SIRT6 is critical for intestinal tuft cell development. Furthermore, many TgSTAT6^{vt} downregulated genes were upregulated by *Sirt6* deficiency, indicating that SIRT6 and STAT6 may be involved in the same pathway to regulate intestinal epithelial homeostasis.

We also thank you for pointing out logic errors for some statements. We have corrected the statement “Mechanistically, SIRT6 ablation induces SOCS3 expression” into “Mechanistically, *Sirt6* ablation causes elevated SOCS3 expression”. In addition, we have carefully read the revised manuscript and made extensive corrections.

4. Key experimental data are missing from the Figure legends; first, while the Methods section states both male and female mice were used, none of the legends specify which sex were used. This is important because of the report that Sirt6 has a sex-specific effect (on lifespan in male mice only). Secondly, while group sizes are given for the experiments presented there is no statement of the number of times each experiment was repeated with similar results.

Response: Although SIRT6 has a sex-specific effect on longevity, we used both male and female mice for the current study, no sex differences were observed. The data we show in the figures are from male mice, we have added the gender information into the figure legends. Moreover, we have added a statement concerning the number of times each experiment was repeated into the “Methods” section.

5. Generally, there are many instances where small differences, or even differences that do not reach statistical significance, are given great weight. For example, in Figure 1 C, D, I or J, none of the data on goblet cells (in or their genes) attain statistical significance. In the same Figure there is total ablation of SIRT6 in KO but really modest effect on tuft cells – in F perhaps 25% reduction. Similarly, in Figure 7B the increment in Sirt6 in the transgenic model is massively higher than increment in P-STAT6, and there are only marginal cellular changes in D, J and K These results imply (a) that SIRT6 is a secondary modifier rather than an inducer; and (b) that its effects may be

dependent on the concentrations of the primary inducers. This could be tested by titrating IL13 in the organoid system.

Response: We agree with the reviewer that SIRT6 mainly functions as a modifier rather than an inducer for most of the cellular processes. Since SIRT6 mainly functions as a histone deacetylase, SIRT6 binds directly to transcription factors and modifies the local chromatin status to regulate gene transcription. In general, the roles SIRT6 plays is somehow dependent on the transcription factor or other co-factors. This is why, in some cases, SIRT6 is required but not sufficient for controlling some cellular events. As shown in the revised Fig. S9, naïve IEC-Tg and WT mice exhibit comparable tuft and goblet cell abundance in the jejunum, confirming the notion that SIRT6 requires other co-factors to control epithelium homeostasis at the steady-state. It is of interest to seek such co-factors in future studies.

6. What is the rationale for sampling the jejunum in all experiments, even those with H.polygyrus – a parasite that inhabits the duodenum?

Response: Since *H.poly* inhabits in the duodenal mucosa, we normally cut the duodenum (about 10-15 cm segment below stomach) segment for worm counting. So, for consistency, we used the jejunum segment (closest to the duodenum) for epithelium phenotype analysis in both naïve and *H.poly*-infected mice. Besides duodenum, other parts of the small intestine, including jejunum and ileum, can sense *H.poly* infection and exhibit tuft and goblet cell hyperplasia in response to activated type 2 immunity.

7. In Figure 8 M-O, the authors show that SOCS3 inhibits P-STAT in two cell lines, but do not use this system to analyze Sirt6; increasing induction of Sirt6 should restore P-STAT6 in a titratable manner.

Response: Thanks for your suggestion. In the revised manuscript, we provided additional WB and luciferase assay experiments to strengthen the mechanism that SIRT6 promotes STAT6 Y641 phosphorylation through inhibiting SOCS3 expression (Fig.8p-u).

Minor Comments

8. *Line 67 secrete.*

Response: Thank you. We have corrected this typo.

9. *Line 116 – is twofold “remarkably reduced”?*

Response: Thank you. We have corrected it to “reduced”.

10. *Line 181 states 150 mM succinate; figure legend (and cited reference) states 100 mM.*

Response: Thanks for your careful review. The succinate concentration we used in the study was 150 mM. We have made corrections in the figure legend.

11. *Figure 4 A. At 9 days post infection, 2 mice are strongly positive, 1 is negative. However the quantification shown in Figure 4B has a very small standard error. How can this be?*

Response: Fig. 4a&b shows the densitometric analysis of the relative abundance of phosphorylated STAT6 normalized to that of total STAT6. The 9 dpi mouse the reviewer mentioned exhibited low expression of both Phospho-STAT6 and total STAT6. So, the normalized value (Phospho-STAT6/total STAT6) is comparable to others in the same group.

12. *Figure 4 C shows significant P-STAT6 in the steady state, not consistent with blot in A that has no staining.*

Response: The very weak bands shown in WB of naïve mice were due to short exposure time. To follow reviewer’s suggestion, we conducted additional immunostaining of P-STAT6 and SIRT6 in the jejunum of naïve mice and mice infected with *H.poly* for different time periods (6 dpi, 9 dpi, 14 dpi) (Fig. 4a&b). The results obtaining from P-STAT6 immunostaining are consistent with those previously described by WB.

13. Figure 4 G The 293T reporter assay is not explained in legend.

Response: We removed the luciferase assay from Figure 4 and put the new luciferase assay data into revised Fig. 8t&u. Legends for Fig. 8t&u were provided accordingly.

14. The Discussion is essentially 3 very long paragraphs, over a page each, which makes it difficult for the reader to follow.

Response: Thanks for your nice comments. We have revised the “Discussion” section.

Reviewer comments, second round review

Reviewer #1 (Remarks to the Author):

The authors have addressed many of the points raised in my first report, which were:

- Accurate description of Sirt6 and Stat6/pStat6 expression during the kinetic of infection
- Rescue experiments with recombinant type-2 cytokine (e.g. IL4/IL13)
- Figure edition (control missing in some of them)
- Reconsideration of the discussion part, following obtention of the new data.

The authors have addressed many of these concerns, performing the requested experiments, and, consequently, obtaining enough data to convincingly answer my questions. The only experiment that has not been done is the one aiming to describe by IHC the total Stat6 expression pattern (although the authors actually tried to perform this staining with 2 different antibodies). Despite that, and given all evidence is now provided, the description of the activation of both Sirt6 and pStat6 following parasitic infection is quite convincing. Regarding the rescue experiment, the authors show that treatment of mice with recombinant IL4/IL13 is unable to rescue the defects of epithelial remodelling observed in a Sirt6-deficiency context, as shown in organoids. Thus, this help to explain how Sirt6 impacts type-2 immune response. Figures and discussion were edited consequently.

In conclusion, my overall feeling is that this manuscript is now suitable for publication.

Reviewer #2 (Remarks to the Author):

The reviewers have addressed my comments.

We appreciate the time and effort that the reviewers dedicated to providing feedback on our manuscript and are grateful for the insightful comments and valuable improvements to our paper.

Reviewer #1 (Remarks to the Author):

The authors have addressed many of the points raised in my first report, which were:

- Accurate description of Sirt6 and Stat6/pStat6 expression during the kinetic of infection
- Rescue experiments with recombinant type-2 cytokine (e.g. IL4/IL13)
- Figure edition (control missing in some of them)
- Reconsideration of the discussion part, following obtention of the new data.

The authors have addressed many of these concerns, performing the requested experiments, and, consequently, obtaining enough data to convincingly answer my questions. The only experiment that has not been done is the one aiming to describe by IHC the total Stat6 expression pattern (although the authors actually tried to perform this staining with 2 different antibodies). Despite that, and given all evidence is now provided, the description of the activation of both Sirt6 and pStat6 following parasitic infection is quite convincing. Regarding the rescue experiment, the authors show that treatment of mice with recombinant IL4/IL13 is unable to rescue the defects of epithelial remodeling observed in a Sirt6-deficiency context, as shown in organoids. Thus, this helps to explain how Sirt6 impacts type-2 immune response. Figures and discussion were edited consequently.

In conclusion, my overall feeling is that this manuscript is now suitable for publication.

Response: Thanks for your comments. We think that the suggestions you have made are very valuable and helpful for revising and improving our paper, as well as the important guiding significance to our researches.

Reviewer #2 (Remarks to the Author):

The authors have addressed my comments.

Response: Thank you.